

# Probabilistic assessment of field-scale CO2 generation by Carbonate/Clay Reactions in sedimentary basins

Giulia Ceriotti[1], Claudio Geloni[2], Matilde Dalla Rosa[2], Alberto Guadagnini[1,3], and Giovanni Porta[1]

[1]Department of Civil and Environmental Engineering, Politecnico di Milano, Piazza L. Da Vinci 32, 20133 Milano, Italy
[2]Eni S.p.A.-Upstream and Technical Services, via Emilia, 1 20097 San Donato Milanese (MI), Italy
[3]Department of Hydrology and Atmospheric Sciences, University of Arizona, Tucson, Arizona, USA

**Correspondence:** Giulia Ceriotti(giulia.ceriotti@polimi.it)

**Abstract.** This work explores the probabilistic assessment of $CO_2$ generation rate and $CO_2$ source location by occurrence of Carbonate/Clay Reactions (CCRs) in three-dimensional realistic sedimentary basins. We ground our assessment on the methodology proposed for a mono-dimensional case study and a single CCR formulation by Ceriotti et al. (2017) which includes a framework to account for thermodynamic parameter uncertainties. This methodology is here extended to a realistic

three-dimensional sedimentary basin case study and transferred to encompass different types of CCRs, including two newly formulated CCRs which accounts for minerals typically observed in sedimentary environments. While testing the ability of the chosen procedure to model diverse CCRs in three-dimensional realistic subsurface sedimentary systems, we quantitatively compare the impact of CCR formulation on the spatial distribution of $CO_2$ source location, temperature and pressure compatible to $CO_2$ gaseous generation, and $CO_2$ generation rate in three-dimensional environments characterized by complex and

non-uniform stratigraphy. Application of the procedure to different types of CCRs enables us to provide an insight on the impact of mineralogical composition on the mechanism of activation temperature and pressure and the amount of $CO_2$ released by the different CCR mechanisms. Finally, we show the implementation of the proposed probabilistic framework to define scenarios associated with different levels of probability to be used as the input and boundary conditions for $CO_2$ migration and transport models in the subsurface.

## 1 Introduction

Carbon dioxide ($CO_2$) is ubiquitously found in gaseous mixtures accumulated across sedimentary basins together with hydrocarbon (e.g., $CH_4$) and other non-hydrocarbon components (e.g., $N_2$; Feng et al., 2016). With reference to gas reservoirs, it might constitute up to 90% of the total gas volume (Wycherley et al., 1999), its presence being often the cause of dilution of

the gas mixture and (sometimes marked) hampering of its energy content (Imbus et al., 1998). Possible sources of such large amounts of $CO_2$ in a sedimentary basin are associated with transformation of organic carbon, carbonate mineral dissolution, inorganic chemical equilibrium of the feldspar-clay-carbonate mineral system, and magma degassing (Smith and Ehrenberg, 1989; Coudrain-Ribstein et al., 1998; Kotarba and Nagao, 2008; Li et al., 2008; Fischer et al., 2006; Ballentine et al., 2001). Processes of $CO_2$ generation and accumulation may be of interest to the characterization of flow and geochemistry in deep





subsurface systems as well as flow in reltively shallow groundwater bodies, as large $CO_2$ accumulations may trigger vertical
flow and transport processes (Kissinger et al., 2017; Marín-Moreno et al., 2019). Characterization and understanding of the key
mechanisms that control the natural formation of carbon dioxide are not completely explored and are still a subject of research.
A considerable body of studies (e.g., Giggenbach, 1980; Smith and Ehrenberg, 1989; Coudrain-Ribstein and Gouze, 1993;
Coudrain-Ribstein et al., 1998; Xu and Pruess, 2001; Cathles and Schoell, 2007; Chiodini et al., 2007; van Berk et al., 2013;
Hutcheon et al., 1990b; Hutcheon and Abercrombie, 1990; Hutcheon et al., 1990a, 1989, 1980, 1993) during the past 40 years
has suggested that the interaction between carbonates, clays/aluminosilicates, and pore water might play the most important
role in controlling $CO_2$ partial pressure in geologic systems. A relevant influence of mineral rock interactions on dissolved
$CO_2$ has been observed also in groundwater and surface water bodies (Shin et al., 2011). Considering a given sedimentary rock
containing carbonates and clays/aluminosilicates, the amount of dissolved $CO_2$ in the pore water is regulated by the chemical
equilibrium among all mineral phases, such a buffering mechanism being typically denoted as Carbonate/Clay Reaction (CCR,
Hutcheon et al., 1990b). The CCR buffering mechanism involves a complex system of geochemical reactions which is typically
condensed in terms of a schematic reaction of the kind

**CCR1:** $5\,Dolomite + Kaolinite + Silica + 2\,H2O$
$$\rightleftharpoons 5\,CO2 + Clinochlore + 5\,Calcite$$

The possibility of occurrence of such a reaction in real (field-scale) sedimentary environments is supported by various stud-
ies (e.g., Coudrain-Ribstein et al., 1998; Hutcheon et al., 1990b; Hutcheon and Abercrombie, 1990). Several representa-
tions/formulations for Carbonate/Clay Reactions that have been proposed share a reaction structure similar to CCR1 and differ
in terms of the carbonate and aluminosilicate phases included therein (Cathles and Schoell, 2007; Coudrain-Ribstein et al.,
1998; Hutcheon et al., 1990b; Zhang et al., 2000). Each of the available Carbonate/Clay Reactions can be characterized by an
equilibrium constant that quantifies the relative partitioning between reactants and products and the amount of $CO_2$ released in
the pore-water when the system attains thermodynamic equilibrium.

Since the Carbonate/Clay buffering system is a reversible process, the CCR mechanism may act either as a $CO_2$ sink or source
depending on local temperature ($T$) and pressure ($P$), these quantities directly controlling the value of the equilibrium constant
associated with the CCRs.

The study of Smith and Ehrenberg (1989) suggests that the equilibrium constant ($K_{CCR1}$) characterizing CCR1 can take values
larger than 1 for temperatures higher than 100 - 120 °C, thus favoring release of $CO_2$ which can then be found as a dissolved
species in pore-water. Starting from this analysis, Cathles and Schoell (2007) propose a simple conceptual model which dis-
tinguishes two possible alternative CCR behaviors. These are exemplified in the depiction of Figure 1 where we consider two
points A and B located at different depths in a sedimentary environment, as described in the following.

– Location A in Figure 1 corresponds to shallow depths. Here, $CO_2$ and all chemical species dissolved in the pore-water
are at equilibrium with the mineral assemblage. $CO_2$ and other gaseous species (e.g., $CH_4$) can appear only as dissolved
phases. The moderate temperature and pressure typically associated with these shallow depths do not promote formation



of large amounts of $CO_2$. Thus, the sum of the partial pressures of all gaseous species attains values that are smaller than the pore-water pressure (i.e., $P_{gas}<P$).

- – Location B corresponds to large depths. High temperature values that are expected to take place at such locations tend
to remarkably shift the equilibrium towards the right-hand side of the CCR reaction, a high amount of $CO_2$ being then released in the pore water. In this scenario, the partial pressure of $CO_2$ (summed to other gaseous compounds including, e.g., aqueous vapor) is typically higher than pore-water pressure (i.e., $P_{gas}>P$). A $CO_2$-rich gaseous phase is then separated and tends to migrate upwards through rock matrix fractures due to buoyancy effects. A disequilibrium between the rock mineral phases and the pore water is then promoted and generation of $CO_2$ takes place until at least
one of the reactants of the CCR system is exhausted. The occurrence of the conditions corresponding to location B is hereafter denoted as CCR mechanism activation, implying that the geochemical disequilibrium and the formation of a separate $CO_2$-rich gaseous phase have been triggered.

Cathles and Schoell (2007) provide a first implementation of the above described conceptual approach upon relying on the linear $T$ - $P$ trend proposed by Smith and Ehrenberg (1989), i.e., $P$[bar] = 6 ($T$ [°C] - 25), and using as a reference three CCR
buffering mineral assemblages, corresponding to $i$) calcite-laumontite-kaolinite-quartz, $ii$) siderite-daphnite-kaolinite-quartz, and $iii$) magnesite-daphnite-kaolinite-quartz. Results of their analysis $a$) suggest that the formation of a separate $CO_2$-rich gaseous phase is feasible for temperature higher than 330°C and $b$) represent the first quantitative estimation of the temperature and pressure of CCR activation as source of gaseous $CO_2$ in a sedimentary environment. However, it should be noted that these results cannot be readily transferred to a generic realistic sedimentary basin scenario because they are associated with
$i$) mineral phases that are rarely observed in real sediments (e.g., laumontite and daphnite), and $ii$) a linear simplified $T$ - $P$ relationship.

Otherwise, $T$ and $P$ evolution in real sedimentary basins often displays complex patterns, each scenario being characterized by site-specific $T$-$P$ spatial and temporal distributions. These are a result of the diagenetic processes of rocks and the burial history of the sedimentary basin itself and should then be appropriately included in a CCR-based assessment of gaseous $CO_2$
generation.

These aspects are fully recognized by Ceriotti et al. (2017) who combine a one-dimensional burial model with a geochemical model formulated according to the conceptual approach suggested in Cathles and Schoell (2007). A key point of novelty introduced by Ceriotti et al. (2017) is the reliance on a probabilistic framework to propagate uncertainty of thermodynamic parameters associated with reaction CCR1 to target modeling goals (i.e., $CO_2$ source location and $CO_2$ generation rate). Such
a stochastic modeling framework allows assessing the probability distribution of $i$) the depth at which the source of gases is located, $ii$) the amount of $CO_2$ generated (conditional to a given mineralogy of the sediments involved in the basin formation process), and $iii$) the range of $T$-$P$ combinations associated with gaseous $CO_2$ generation.

In this context, an appraisal of this probabilistic approach considering a fully three-dimensional scenario with the ensuing quantification of the amount of $CO_2$ that can be realistically released by CCR reactions is still lacking. This is precisely the
key goal of this study, which is geared to $i$) estimating the spatial distributions of $CO_2$ sources and the associated generation





rates in realistic three-dimensional sedimentary basins and $ii$) assessing differences in the activation temperature and pressure characterizing various possible formualtions of the CCR mechanism. We remark that the evaluation of all these quantities is still a major element of study and debate in the literature (Jarvie and Jarvie, 2007). To this end, we start from the methodology proposed in Ceriotti et al. (2017) and explore the transferability of this probabilistic approach to a realistic three-dimensional

sedimentary basin setting. We rest on reaction CCR1 as a test bed for the CCR mechanism. To explore the impact of CCR formulations on model results, we consider various types of CCRs, each characterized by uncertain thermodynamic parameters. We then provide a critical discussion of the activation of the CCR mechanism linked to these different CCR formulations upon examining the corresponding probability distributions of the $CO_2$ source location, as well as of the activation temperature and pressure. We finally discuss the implications of relying on such approach to delineate alternative scenarios, each associated

with a given level of probability, which may feed models describing $CO_2$ migration and transport in the subsurface.

The work is structured as follows. Section 2 is devoted to the presentation of the three-dimensional sedimentary setting and of the CCR formulations we consider. These include a typically employed formulation and two additional models involving clay and silicate minerals (such as beidellite and illite) that are frequently observed in sedimentary basins. Section 3 summarizes the modeling and uncertainty quantification workflow and procedures employed. Results are presented and discussed in Section 4.

Finally, concluding remarks are provided in Section 5.

## 2   Sedimentary setting and CCR formulations

The reference system considered in this study is a three-dimensional realistic sedimentary basin with a deposition history spanning a temporal window of 135 Ma (Millions of years before present) and characterized by the deposition sequence listed in Table 1.

| Layer | Time interval of deposition | Density, $\rho$ [kg$_{sed}$ m$_{sed}^{-3}$] |
|---|---|---|
| Carbonate 1 | from 135 Ma to 94 Ma | 2600 |
| Carbonate 2 | from 94 Ma to 48 Ma | 2600 |
| Carbonate 3 | from 48 Ma to 34 Ma | 2600 |
| Carbonate 4 | from 34 Ma to 23 Ma | 2670 |
| Shale 1 | from 23 Ma to 5.3 Ma | 2500 |
| Shale 2 | from 5.3 Ma to 0 Ma | 2600 |

**Table 1.** Sequence of sediments deposited during the 135 Ma of basin deposition history and sediment density.

The basin stratigraphy at the present time (which is taken as $t = 0$ Ma) is depicted in Figure 2 and comprises six layers (corresponding to four carbonate and two shale rock systems). The planar surface of the basin covers an area of about 177.5 km $\times$ 155 km, the maximum depth (below sea level) reached by the volume filled by sediments being approximately 8 km. The geo-history of the basin is reconstructed using the widely tested and documented burial model E-SIMBA™ (for details see, e.g., Grigo et al., 1993; Dalla Rosa et al., 2015; Zattin et al., 2016) which allows estimating the three-dimensional dynamic





evolution of stratigraphy as well as temperature, pressure and porosity distributions. These variables are here taken as input data.

Figure 3 depicts the spatial distribution of temperature ($T$ [°C] in panel A) and pressure ($P$ [bar] in panel B) along two perpendicular vertical cross-sections located at $x = 32$ km and $y = 105$ km, respectively (see the reference system indicated in Figures 2-3). Note that the $z$-axis points downwards, i.e., the value of $z$ increases with depth. Each cell of the spatial mesh

used to describe the evolution of $T$, $P$ and porosity has a uniform size of 2500 m $\times$ 2500 m $\times$ 200 m ($x \times y \times z$). Temperature and pressure display an overall increasing trend with depth which yields values of $T$ and $P$ close to those typically observed in real sedimentary basins (e.g., Colombo et al., 2017, 2018). The largest temperature and pressure values (corresponding to 330 °C and 800 bar, respectively) are observed at the deepest locations in the basin.

Considering the above described reference geological setting, we investigate separately three differing CCR formulations

which can be considered at the basis of $CO_2$ generation. These include the classical reaction CCR1 (illustrated in Section 1 and recalled in the following) along with two alternative CCR models (labeled CCR2 and CCR3):

**CCR1:** $5\,Dolomite + Kaolinite + Silica + 2\,H2O$
$$\rightleftharpoons 5\,CO2 + Clinochlore + 5\,Calcite$$

**CCR2:** $0.33\,Dolomite + 1.13\,Microcline + Beidellite + 0.33\,H2O$
$$\rightleftharpoons 0.33\,CO2 + 1.33\,Illite + 1.5\,Quartz + 0.33\,Albite + 0.33\,Calcite$$

**CCR3:** $0.33\,Dolomite + 1.13\,Microcline + Beidellite + 0.66\,H2O$
$$\rightleftharpoons 0.33\,CO2 + 1.33\,Illite + 1.8\,Quartz + 0.33\,Analcime + 0.33\,Calcite$$


CCR2 and CCR3 are here proposed based on laboratory tests aimed at investigating the role of different types of clay in sedimentary environments (Panariti and Previde Massara, 2000). These formulations include mineral phases (such as beidellite, analcime and microcline) which can be considered a proxy of clays and feldspars that have been observed promoting the release of $CO_2$ by dolomite in laboratory experiments. The ability of CCR2 and CCR3 to interpret field $P_{CO_2}$ data is further discussed

in Section 3.2.

Depending on the CCR investigated, we consider a given mineralogical composition of sediments, as listed in Table 2. These mineralogies (termed as M1 for CCR1 and M2-3 for CCR2 and CCR3) are selected to maximize the mass of $CO_2$ that can potentially be generated by a unit mass of sediment ($m_{CO_2}$ [$kg_{CO_2}\,kg_{sed}^{-1}$]) when a prescribed CCR mechanism is activated. Details on the computation of $m_{CO_2}$ are reported in the Supplementary Material of Ceriotti et al. (2017). For simplicity, we

assume here that the four carbonate rocks forming the sedimentary basin described in Figure 2 are characterized by the same uniformly distributed mineralogical composition, i.e., M1 or M2-3 when modeling $CO_2$ generation by CCR1 or CCR2 and CCR3, respectively. Otherwise, the shale rocks are assumed to be characterized by a negligible carbonate content, being then





| Label | CCR | Composition (weight/weight %) | $m_{CO_2}$ |
|-------|-----|-------------------------------|------------|
| M1 | CCR1 | Dolomite = 76 % | 0.182 $\text{kg}_{CO_2}$ $\text{kg}_{sed}^{-1}$ |
| | | Kaolinite = 19 % | |
| | | Silica = 5 % | |
| M2-3 | CCR2 | Dolomite = 8 % | 0.020 $\text{kg}_{CO_2}$ $\text{kg}_{sed}^{-1}$ |
| | CCR3 | Microcline = 42 % | |
| | | Beidellite = 50 % | |

**Table 2.** Composition of the mineralogical scenarios used for the investigation of the three CCRs considered. The mass of $CO_2$ released by a unit mass of sediment ($m_{CO_2}$ [$\text{kg}_{CO_2}$ $\text{kg}_{sed}^{-1}$]) when the gaseous $CO_2$ generation is activated is also listed for each setting.

incompatible with the occurrence of CCR (i.e., we assume that $m_{CO_2}$ associated with shale layers is zero). We emphasize that the proposed methodological framework and modeling approach are fully compatible with the presence of a spatially variable

mineralogical composition. As such, our approach can be employed to assess the impact of uncertainties associated with spatially heterogeneous arrangements of mineral and sediment composition on CCR-based $CO_2$ generation. The latter could be tackled upon relying on appropriate techniques such as, e.g., Functional Compositional Kriging (see, e.g., Menafoglio et al., 2016, and references therein). Analyzing this aspect is, however, beyond the scope of the present study and will be pursued in future works.

**3   CCR modeling under uncertainties**

The modeling approach used in this work relies on the assumption that a CCR (such as CCR1, CCR2 and CCR3 (see Section 2)) involves only pure phases and can be destructured as a sum of speciation reactions. Defining the speciation reactions and their equilibrium constant is the first step required to compute the amount of $CO_2$ generated by CCRs and to identify the $CO_2$ source location. We follow Ceriotti et al. (2017) and consider the equilibrium constant of speciation reactions as the key source of

uncertainty that is propagated throughout the final modeling goals of interest, i.e., the $CO_2$ source location, the $CO_2$ generation rate, and the temperature and pressure of CCR activation. Here, we ($a$) briefly recall the concept of speciation reaction and and methodology employed to quantify the uncertainty associated with the speciation equilibrium parameter (Section 3.1); and ($b$) illustrate the approach leading to the computation of $CO_2$ partial pressure (Section 3.2) and to the identification of $CO_2$ source and generation rate (Sections 3.3 and 3.4, respectively). Additional details on the methodologies employed can be found in

Ceriotti et al. (2017).

**3.1   Speciation reactions and uncertainty characterization**

Given a generic mineral, aqueous or gaseous phase ($Ph$), it always possible to describe the speciation in water of phase $Ph$ upon relying on a set of aqueous basis species (Anderson, 2005). A speciation reaction can then be characterized by an equilibrium constant ($K_{S,Ph}$), whose value depends on the system temperature and pressure. Following Ceriotti et al. (2017),





we assume that the equilibrium constant driving speciation of $Ph$ can be expressed as

$$\log \widetilde{K}_{S,Ph} = \widetilde{A} + B \cdot T + \frac{C}{T} + \widetilde{D} \cdot \log T + \frac{E}{T^2} \tag{1}$$

where $T$ [K] is temperature and the symbol $\tilde{}$ denotes (uncertain) random variables (to distinguish these from deterministic quantities). Note that this formualtion holds for a given pressure of $P$ = 1 bar. The format of Eq. (1) resembles the one characterizing the expression of temperature dependent equilibrium constant derived from the Maier-Kelley heat capacity

assumption (Maier and Kelley, 1932) which is typically used in thermodynamic databases (Parkhurst and Appelo, 2013; Blanc et al., 2012; Delany and Lundeen, 1990). The key difference between Eq. (1) and the classical expression for (temperature dependent) equilibrium constant is that the two parameters $\widetilde{A}$ and $\widetilde{D}$ are not considered as deterministic effective parameters and are here interpreted as bivariate Gaussian random variables. We follow the approach of Ceriotti et al. (2017) to define the mean values ($\mu_A$ and $\mu_D$ for $\widetilde{A}$ and $\widetilde{D}$, respectively) and the entries of the covariance matrix $\Psi$ characterizing the bivariate Gaussian

distribution of $\widetilde{A}$ and $\widetilde{D}$. Given the structure of Eq. (1), it then follows that, for a given temperature value $log\widetilde{K}_{S,Ph}$ is described by a normal distribution with parameters related to the statistical moments of $\widetilde{A}$ and $\widetilde{D}$. Details about the characterization of $\widetilde{A}$ and $\widetilde{D}$ for all phases appearing in this study are reported in the Supplementary Material. Uncertainties associated with the characterization of $\widetilde{A}$ and $\widetilde{D}$ can be propagated to the $Ph$ speciation equilibrium constant through Eq. (1). It then follows that $\widetilde{K}_{S,Ph}$ is not a deterministic quantity but rather an uncertain variable described by a probability density function ($pdf$).

**3.2 CO$_2$ partial pressure computation**

We introduce here a generalized CCR formulation in the form of

**CCR:** $\alpha_1\, Ph_1 + ... + \alpha_i\, Ph_i + ... + \alpha_I\, Ph_I$
$$\rightleftharpoons \alpha_{I+1}Ph_{I+1} + ... + \alpha_{I+J}\, Ph_{I+J} + \alpha_0 CO_2$$

where the symbol $Ph_i$ indicates the $i^{th}$ phase (with $i = 1...I+J$) appearing in the CCR, the term $\alpha_i$ representing the stoichiometric coefficient of phase $i$; $I$ and $J$ quantify the number of CCR reactants and products, respectively, with the exception of

$CO_2$ which is explicitly accounted on the right-hand side of the CCR with its stoichiometric coefficient, $\alpha_0$. Each of the phases involved in the CCR is associated with a speciation reaction and an uncertain speciation equilibrium constant, as described in Section 3.1. Note that the proposed generic CCR formulation can be readily recast into CCR1, CCR2, or CCR3.

We can express the equilibrium constant of the CCR ($\widetilde{K}_{CRR}$) as (Anderson, 2005)

$$\log \widetilde{K}_{CCR}(T) = \sum_{i=1}^{I} \alpha_i \log \widetilde{K}_{S,Ph_i} - \sum_{i=I+1}^{J+I} \alpha_i \log \widetilde{K}_{S,Ph_i} - \alpha_0 \log \widetilde{K}_{S,CO_2} \tag{2}$$

The quantities $\widetilde{K}_{S,CO_2}$ and $\widetilde{K}_{S,Ph_i}$ correspond to the speciation equilibrium constants associated with $CO_2$ and the $i^{th}$ phase contributing to the CCR, respectively. The uncertain variables $\widetilde{K}_{S,Ph_i}$ and $\widetilde{K}_{S,CO_2}$ are evaluated through Eq. (1) as a function of temperature. The value of $\widetilde{K}_{CCR}$ resulting from Eq. (2) is then temperature dependent and affected by uncertainty. The





effect of pressure on $\widetilde{K}_{CCR}$ is considered through a correction term (Millero, 1982)

$$\log \widetilde{K}_{CCR}(T,P) = \log \widetilde{K}_{CCR}(T, P = 1) - \frac{\Delta V^\circ}{2.303 R_g T} \cdot (P - 1) \tag{3}$$

where $\widetilde{K}_{CCR}(T,P)$ is the CCR equilibrium constant computed for a generic value of $T$ and $P$; $\widetilde{K}_{CCR}(T, P = 1)$ is the CCR equilibrium constant computed for a generic value of $T$ and pressure $P$ = 1 bar as resulting from Eq. (2); $\Delta V^\circ$ [m$^3$ mol$^{-1}$] represents the change of the molar volume associated with the CCR; and $R_g$ is the ideal gas constant.

The partial pressure of $CO_2$ ($\widetilde{P}_{CO_2}$) associated with the CCR can then be evaluated as (Coudrain-Ribstein et al., 1998; Cathles

and Schoell, 2007; Ceriotti et al., 2017)

$$\log \widetilde{P}_{CO_2}(P,T) = \frac{\log \widetilde{K}_{CCR}(P,T)}{\alpha_0} \tag{4}$$

Equation (4) rests on the assumption that the $CO_2$ fugacity coefficient is set to unity (Hutcheon et al., 1990b; Chiodini et al., 2007; Ceriotti et al., 2017). Equations (2) - (4) allow computing the partial pressure of $CO_2$ as a function of basin temperature and pressure, yielding a three-dimensional distribution of $\widetilde{P}_{CO_2}$ as a function of basin stratigraphy and burial history. To provide

a preliminary assessment, Figure 4 reports the mean values of $\log \widetilde{P}_{CO_2}$ associated with CCR1, CCR2 and CCR3 as a function of temperature, assuming that $P$ [bar] = 6×($T$ [° C] - 22) (Smith and Ehrenberg, 1989). The $\log \widetilde{P}_{CO_2}$ trends are compared against values of $P_{CO_2}$ sampled in different sedimentary basins obtained from literature (Coudrain-Ribstein et al., 1998). We note that, on the one hand, mean $\log \widetilde{P}_{CO_2}$ trend associated with CCR1 provides a good interpretation of data observed at temperatures larger than 100 °C (specifically for Norway, Texas and Thailandia basins). On the other hand, $\log \widetilde{P}_{CO_2}$ mean

trend resulting from CCR2 and CCR3 formualtions appears to explain data observed at lower temperatures, ranging between 50 and 100 °C (Norway, Paris Basin and Arkansas). This is consistent with the considerations already provided by Coudrain-Ribstein et al. (1998) who suggest that CCR formualtions accounting for complex clay phases (such as illite) can feasibly interpret low-temperature $P_{CO_2}$ trends. We can conclude that the three formulations considered in this work are compatible with data observed in real sedimentary environments.

**3.3  CO$_2$ source localization**

According to the conceptual model of Cathles and Schoell (2007), the CCR mechanism activates when the sum of the partial pressures of all gaseous species is higher than the pore-water pressure (see Section 1). Here, we assume that only $CO_2$ and aqueous vapor partial pressure might contribute to the formation of a $CO_2$-rich separate gas phase while the effect of other gas species (e.g., hydrocarbon gases) is neglected.

For a selected observation time ($t = \widehat{t}$) and location (identified by the coordinates $x = \widehat{x}$ and $y = \widehat{y}$) on the planar surface of the sedimentary basin, we define the quantity $\widetilde{R}(\widehat{t}, \widehat{x}, \widehat{y}, z)$ as

$$\widetilde{R}(\widehat{t}, \widehat{x}, \widehat{y}, z) = \frac{\widetilde{P}_{CO_2}(\widehat{t}, \widehat{x}, \widehat{y}, z) + P_v(\widehat{t}, \widehat{x}, \widehat{y}, z)}{P(\widehat{t}, , \widehat{x}, \widehat{y}, z)} \tag{5}$$





Here, the symbol $P_v$ denotes the aqueous vapor partial pressure, which we evaluate according to the procedure described by Ceriotti et al. (2017). The variable $\widetilde{R}$ is affected by uncertainty because it depends on the random variable $\widetilde{P}_{CO_2}$, on elevation $z$,

and can undertake values equal to or larger than unity when a location is compatible with the activation of a CCR mechanism. The $CO_2$ source ($\widetilde{Z}_{act}(\widehat{t}, \widehat{x}, \widehat{y})$) is then evaluated as the position corresponding to the shallowest vertical coordinate $z$ where $\widetilde{R}(\widehat{t}, \widehat{x}, \widehat{y}, z) \geq 1$. Application of this procedure for all combinations of $x$ and $y$ coordinates enables us to delineate a CCR activation surface in the three-dimensional basin as the collection of points with coordinates $(x, y, z)$ with $z = \widetilde{Z}_{act}(x, y)$, i.e., where the CCR mechanism is activated.

### 3.4   $CO_2$ generation rate

We provide an estimate of the rate of $CO_2$ generated by the CCR mechanism activation per unit area of the CCR activation surface ($\widetilde{F}_{CO_2}(t, x, y)$, [$\mathrm{kg}_{CO_2}$ $\mathrm{Ma}^{-1}$ $\mathrm{m}^{-1}$]) as

$$\widetilde{F}_{CO_2}(t, x, y) = m_{CO_2} \cdot [1 - \phi] \cdot v_b \cdot \rho \tag{6}$$

where $m_{CO_2}$ [$\mathrm{kg}_{CO_2}$ $\mathrm{kg}_{sed}$] is the mass of $CO_2$ released by a unit mass of sediment upon activation of CCR, which depends on

the CCR formulation and mineral composition (see Section 2); $\phi$ [-] and $\rho$ [$\mathrm{kg}_{sed}$ $\mathrm{m}_{sed}^{-3}$] are the sediment porosity and density, respectively; and $v_b$ [m $\mathrm{Ma}^{-1}$] is the burial velocity of sediments, a quantity governing the rate at which the sediments reach the location of the source. As opposed to porosity, the density of a given sediment type can be taken as a constant, its value being listed reported in Table 1 for each type of rock. The quantity $\widetilde{F}_{CO_2}(t, x, y)$ depends indirectly on the activation depth $\widetilde{Z}_{act}(x, y)$ since both $\phi$ and $v_b$ are a function of $z$, their value in Eq.(6) being related to the depth of the $CO_2$ source. Outputs of

the burial model employed in this study (i.e., E-SIMBA$^{\mathrm{TM}}$, see Section 2) do not include the space-time evolution of $v_b$ across the basin. Results from a series of preliminary investigations (not shown here) performed with a one-dimensional burial model (STREAM, see, e.g., Formaggia et al., 2013) at various planar locations of the three-dimensional basin investigated suggest that the burial velocity of sediments does not significantly vary along depth for $z > 2$ km, where the CCR activation is more likely to occur. The value of $v_b$ in these regions is approximately equal to 40 m $\mathrm{Ma}^{-1}$. We take this as a representative value

for $v_b$ in Eq. (6) in our analyses, thus disregarding the vertical variation of burial velocity.

## 4   Results and Discussion

We perform the probabilistic assessment of the CCRs introduced in Section 1 upon relying on a numerical Monte Carlo (MC) approach. Parameters $\widetilde{A}$ and $\widetilde{D}$ associated with each phase $Ph_i$ appearing in a given CCR are sampled $N$ times (for a total of $N = 10^5$ Monte Carlo replicates for each CCR mechanism) to yield $N$ arrays

$$V_n = \begin{vmatrix} A_1 & \cdots & A_i & \cdots & A_{I+J} \\ D_1 & \cdots & D_i & \cdots & D_{I+J} \end{vmatrix}$$

where $V_n$ indicates the $n^{th}$ sampled array (with $n$= 1, ..., $N$) and quantities $A_i$ and $D_i$ represent the $n^{th}$ values sampled from the bivariate Gaussian distribution of $\widetilde{A}$ and $\widetilde{D}$ associated with the $i^{th}$ phase (i.e., $Ph_i$) appearing in the generalized CCR





formulation (see Section 3.2). The modeling approach detailed in Section 3 is applied for each sample $V_n$ to yield $N$ MC realizations of the CCR mechanism occurrence as a function of space and time. The results presented and discussed in this
Section are all associated with $t = 0$ Ma, i.e., the present time, corresponding to the setting when the basin structure reaches the largest depths and the highest temperature and pressure are observed (see Figure 3, Section 2). Note that the modeling approach can be applied to any time level across the basin burial history.

## 4.1 Source location, activation temperature and pressure

By relying on the $N$ MC realizations of our model, we compute the frequency at which the activation of the CCR mechanism
is observed at each location in the sedimentary space ($C_A(x,y,z)$). We start by focusing on the quantity

$$f(\widetilde{Z}_{act}) = \frac{C_A(x,y,z)}{N} \tag{7}$$

which quantifies the three-dimensional distribution of the relative frequency of source location.

Figure 5 displays $f(\widetilde{Z}_{act})$ for CCR1 (A), CCR2 (B), and CCR3 (C) evaluated at $t = 0$ Ma using Eq. (7) along the two cross-sections of the basin depicted in Figure 3. While the three CCRs analyzed yield similar qualitative patterns of $f(\widetilde{Z}_{act})$, some
key quantitative differences can be noted. The spatial region associated with non-zero probability to observe activation of the CCR mechanism (i.e., $f(\widetilde{Z}_{act}) > 0$) is broadest for CCR2. Moreover, Figure 5 suggests that values of $f(\widetilde{Z}_{act})$ do not increase monotonically with depth and attain their largest values at different depths, depending on the considered CCR. These maximum values are located approximately at $\simeq 7$ km for CCR1, at depths ranging between 5 km and 6 km for CCR2, and at $\simeq 6.5$ km for CCR3. The documented peak in $f(\widetilde{Z}_{act})$ and the ensuing decreasing trend observed for very large depths is consistent with
the assumptions underlying our conceptual model, according to which the $CO_2$ source is positioned in the shallowest point where a combination of temperature and pressure compatible with $CO_2$ generation is first attained.

Further to this, our results show that the three CCRs examined yield markedly different ranges of values of $f(\widetilde{Z}_{act})$, the largest observed value for CCR1 being 0.1 (i.e., the probability of activation of CCR1 at given location can be as high as 10%), while being considerably lower for CCR2 and CCR3 (corresponding to 5% and 3%, respectively). $CO_2$ generation by CCR1
is associated with a high frequency in the thin layer of sediment located at $\simeq 7$ km depth. Otherwise, CCR2 and CCR3 display a smooth spatial distribution of $f(\widetilde{Z}_{act})$, displaying a smaller maximum value of $f(\widetilde{Z}_{act})$ if compared with CCR1.

The differences observed in $f(\widetilde{Z}_{act})$ indicate that $(i)$ the $CO_2$ generation occurrence is sensitive to the selected buffering CCR mechanism and $(ii)$ relevant shifts in the source location, characteristic temperature and pressure of activation may be expected as a function of the CCR considered. This element is further explored through the analysis of the probability densitiy function
($pdfs$) of $\widetilde{T}_{act}$ and $\widetilde{P}_{act}$ and their comparison against the $pdf$ of $\widetilde{Z}_{act}$. The latter is evaluated as

$$pdf(\widetilde{Z}_{act}) = \frac{\int_x \int_y C_A(x,y,z)dxdy}{\int_x \int_y \int_z C_A(x,y,z)dxdydz} \tag{8}$$

The $pdfs$ of temperature ($\widetilde{T}_{act}$) and pressure of activation ($\widetilde{P}_{act}$) of the CCR mechanism are evaluated from the three-dimensional distribution of $C_A(x,y,z)$ and the temperature and pressure computed in the burial basin model.





Figure 6 depicts the sample $pdfs$ obtained for CCR1, CCR2, and CCR3. While these $pdfs$ are characterized by a seemingly
similar shape, each of them embeds the signature of the corresponding CCR mechanism, as seen in terms of spread, mean,
and mode (see also Table 3). For example, the mean and mode of the activation temperature are lowest for CCR2, the highest
values being associated with CCR1. This can be explained upon observing that the mean of $\log\widetilde{K}_{CCR2}$ is more sensitive to
temperature than $log\widetilde{K}_{CCR1}$ and $\log\widetilde{K}_{CCR3}$ (see Figure 4). This implies that, on average, CCR2 is activated at lower tem-
peratures than CCR1 and CCR3. For the three considered CCRs, the mean temperature of activation is comprised between
$246°C$ and $287°C$, values which are significantly lower than the threshold of $330°C$ reported by Cathles and Schoell (2007).
The standard deviation $(\sigma)$ of the distribution of $\widetilde{T}_{act}$ depends on the CCR mechanism considered (Table 3). The combination
of higher spread and lower mean characterizing the sample $pdf$ of $\widetilde{T}_{act}$ for CCR2 yields non-zero probability values even for
quite low values of temperature (i.e., $159°C$, see Table 3) as compared to the results of the preliminary assessments of Cathles
and Schoell (2007).

Similar observations can be drawn from the sample $pdf$ of $\widetilde{P}_{act}$ depicted in Figure 6B. While pressure is known to have a
limited impact on equilibrium constants of reactions, our results reveal its major role in the activation of the CCR mechanism.
This is related to the observation that pore-water pressure sets the threshold that is required to be exceeded so that a separate
gas phase can be found in the system. As such, the key statistics of $\widetilde{P}_{act}$ depend on the CCR mechanism investigated (Table 3).
Figure 6 depicts the sample $pdf$ of $\widetilde{Z}_{act}$ for the three CCR mechanisms analyzed. The behavior of these results mirrors the one
displayed by the distributions of $\widetilde{T}_{act}$ in Figure 6A. According to our probabilistic assessment, the distribution of $\widetilde{Z}_{act}$ and the
associated main statistics (Table 3) suggest that CCR2 is the activation mechanism which tends to take place at the shallowest
depths. Indeed, while the mode and the mean of $\widetilde{Z}_{act}$ are larger than 6.60 km for CCR1 and CCR3, the source depth with high-
est probability is found at about 5.78 km for CCR2. A similar behavior is shown for the mean of of $\widetilde{Z}_{act}$. The higher spread
associated with the population of sampled $\widetilde{T}_{act}$ values for CCR2 is mirrored by the behavior of $\widetilde{Z}_{act}$. As a consequence, the
shallowest depth where $CO_2$ generation might take place under the action of CCR2 corresponds to 3.2 km from the sea level,
which is about 1.4 km smaller than that observed for CCR1. Note that the distributions of $\widetilde{T}_{act}$ and $\widetilde{P}_{act}$ collected in Figure 6
provide a first quantitative assessment of the temperature and pressure of activation of $CO_2$ generation characterizing CCR1,
CCR2, and CCR3. Thus, results of this kind can be used to perform preliminary probabilistic evaluations of CCR activation as
a $CO_2$ source.

The extent of the impact of the CCR formulations considered on the occurrence of $CO_2$ generation can also be assessed by
analyzing the relative frequency of activation associated with each point of the basin planar surface ($f_A (x,y)$). The latter
isdepicted in Figure 7A and has been estimated as

$$f_A(x,y) = \sum_{z=0}^{z=Z_T(x,y)} f(\widetilde{Z}_{act}) \tag{9}$$

where $Z_T$ is the maximum depth attained for each pair of coordinates $(x, y)$ in the basin at $t = 0$ Ma. Figures 7B, C and D
depict the spatial distribution of $f_A(x,y)$ for CCR1, CCR2, and CCR3, respectively. These results indicate that the frequency
of activation of the CCR mechanism is spatially heterogeneous. Its distribution shows a pattern that is closely dependent on
the maximum depth attained by the sediments (see Figure 7), being linked to the burial history of the considered basin. For





| CCR | $\mu$ | $\sigma$ | mode | min |
|---|---|---|---|---|
| | | $\widetilde{T}_{act}$ | | |
| CCR1 | 287°C | 21°C | 281°C | 220°C |
| CCR2 | 246°C | 31°C | 237°C | 159°C |
| CCR3 | 273°C | 30°C | 280°C | 185°C |
| | | $\widetilde{P}_{act}$ | | |
| CCR1 | 764 bar | 26 bar | 771 bar | 650 bar |
| CCR2 | 716 bar | 39 bar | 691 bar | 569 bar |
| CCR3 | 748 bar | 36 bar | 751 bar | 610 bar |
| | | $\widetilde{Z}_{act}$ | | |
| CCR1 | 6.78 km | 0.561 km | 6.80 km | 4.6 km |
| CCR2 | 5.78 km | 0.832 km | 5.40 km | 3.2 km |
| CCR3 | 6.46 km | 0.778 km | 6.60 km | 4.0 km |

**Table 3.** Mean ($\mu$), standard deviation ($\sigma$), mode and minimum value associated with the sample $pdfs$ of $\widetilde{T}_{act}$, $\widetilde{P}_{act}$ and $\widetilde{Z}_{act}$. Statistics are computed for CCR1, CCR2 and CCR3. The maximum values of $\widetilde{T}_{act}$, $\widetilde{P}_{act}$ and $\widetilde{Z}_{act}$ are not reported as they correspond to the maximum temperature, pressure and depth observed in the selected settind independently from the target CCR mechanism.

all CCRs explored, the highest relative frequency of activation is observed at a location $x$ and $y$ where the basin stratigraphy is the thickest (i.e., $\simeq$ 8 km in our setting). This is consistent with the observation that sediments reaching deeper locations

experience higher temperatures, thus leading to an overall increase of the probability that activation of CCR be observed for a given location $(x, y)$. We note that both CCR1 and CCR2 are characterized by a maximum value of $f_A(x,y)$ equal to 0.7, i.e., there is a planar location in the system where activation of these CCRs along some vertical takes place in 70% of the $N$ MC realizations. On the other hand, the largest values of $f_A(x,y)$ for CCR3 attain values that are about 0.3, i.e., significantly smaller than those recordered for CCR1. This result is consistend with the observation that CC3 is less likely to activate than

CCR1 at large depths, as suggested by the spatial distributions of $f(\widetilde{Z}_{act})$ reported in Figure 5A and C.

Our probabilistic assessment documents that the characteristic temperature and pressure associated with the activation of the CCR mechanism are driven by $(a)$ the considered CCR formulation and $(b)$ the mineralogical assemblage constituting the buffering systems. Thus, the risk of $CO_2$ generation taking place at some depth in a sedimentary basin is markedly dependent on the three-dimensional temperature and pressure distribution as well as the selected buffering system.

The probabilistic delineation of the source location may profoundly depend on the CCR mechanism employed in the modeling workflow. This result is of key relevance in light of a subsequent analysis involving modeling of transport, migration and accumulation of the generated $CO_2$. Shallow sources are typically associated with a reduced traveling path of gaseous $CO_2$ and a decreased possibility of $CO_2$ re-mineralization. Therefore, a location of the $CO_2$ source at relatively shallow depths may increase the risk of observing large accumulation in reservoirs of interest for oil and gas exploration, as well as the probability





that $CO_2$ migration may influence vertical flow processes capable of influencing shallow groundwater bodies.

## 4.2 Implications for a scenario-based $CO_2$ migration modeling

When dealing with subsurface $CO_2$ migration modeling, a key step is the design of the input scenario, i.e., the definition of a location of the $CO_2$ source (i.e., activation surface in a three-dimensional setting) and the $CO_2$ generation rate. Our probabilistic framework can assist the design of multiple scenarios. In practice, this can be obtained through the followign steps:

1. the solution of Eqs. (1)-(4) for all $N$ Monte Carlo samples yields the $pdf$ characterizing $\widetilde{P}_{CO_2}$ at each spatial location of the three-dimensional sedimentary basin, such a $pdf$ being conditional to the $T$ and $P$ values rendered by the burial model.

2. starting from the cumulative probability distribution of $\widetilde{P}_{CO_2}$ we obtain scenarios of $CO_2$ partial pressure ($pp_w(CO_2)$), each associated with a given percentile ($p_w$).

3. a given scenario $pp_w(CO_2)$ constitutes the input to the system of Eqs. (5) - (6) for the evaluation of the spatial distribution of $Z_{act}$ and $F_{CO_2}$ associated with the percentile (or probability level) $w$.

For the considered time $t = 0$ Ma, we exemplify the types of scenarios which can be used as input for $CO_2$ transport modeling by $(a)$ selecting the $25^{th}$, $50^{th}$, $75^{th}$ and $99^{th}$ percentiles of the sample $pdf$ of $\widetilde{P}_{CO_2}$ at each point in the sedimentary basin and $(b)$ building corresponding three-dimensional scenarios of $CO_2$ partial pressure distribution ($pp_w(CO_2)$, with $w = 25, 50,$ 75, 99) for each of the CCRs investigated in this study.

Figure 8 depicts the spatial location of the activation source associated with the $50^{th}$, $75^{th}$ and $99^{th}$ percentile of the distributions stemming from CCR1 (A), CCR2 (B), and CCR3 (C). Note that, regardless the selected CCR formulation, when considering the $25^{th}$ percentile of the $CO_2$ partial pressure $pdfs$ (corresponding to the $pp_{25}(CO_2)$ scenarios) none of the locations in the basin satisfies the criterion of CCR mechanism activation. Thus, an activation surface is not observed for $pp_{25}(CO_2)$ scenarios. The same reason motivates the lack of activation surfaces associated with $pp_{50}(CO_2)$ and $pp_{75}(CO_2)$ for CCR3 in Figure 8C.

Comparison of Figure 8A and Figure 8B indicates that the scenarios corresponding to the $50^{th}$ and $75^{th}$ percentiles yield activation surfaces with similar extent and average depth for CCR1 and CCR2. Otherwise, the scenario associated with the $99^{th}$ percentile displays markedly different features across the CCRs analyzed, the activation surface characterizing CCR2 being located at considerably shallower depths (and hence being more extended) than its counterparts in CCR1 or CCR3 (Figure 8). This result descends from the differences between the CCRs observed for the probability densities of the activation mechanism at relatively low temperature, i.e., $T_{act} < 250°$C, as discussed in Section 4.1 and illustrated in Figure 6. Therefore, the extent and location of the activation surface may deeply change, depending on the selected CCR, as well as its characteristic activation temperature. This aspect is further elucidated in the detailed depiction of Figure 9 which juxtaposes the activation surfaces associated with $pp_{99}(CO_2)$ for CCR1 and CCR2, the color scale quantifying the $CO_2$ generation rate per unit square meter



($F_{CO_2}$ [kg $CO_2$ m$^{-2}$ Ma$^{-1}$]). While CCR2 yields an activation surface with a larger spatial extent, CCR1 is characterized by a higher specific $CO_2$ generation rate. Values of mean $\mu(F_{CO_2})$ and standard deviation $\sigma(F_{CO_2})$ of $F_{CO_2}$ observed for both activation surfaces displayed in Figure 9 are listed in Table 4. These results show that $\mu(F_{CO_2})$ is almost one order of

magnitude larger for the CCR1 activation surface than for CCR2. This is consistent with the values of $m_{CO_2}$ associated with CCR1 (0.182 kg$_{CO_2}$ kg$_{sed}^{-1}$) and for CCR2 (0.02 kg$_{CO_2}$ kg$_{sed}^{-1}$) (see Table 2). Because we assume here a constant burial velocity in Eq. (6), $m_{CO_2}$ is the main quantity affecting $F_{CO_2}$ which varies mildly across the activation surface for both CCRs (see values of standard deviations in Table 4), a result which is in line with the modest spatial variability of porosity resulting from the burial model.

The overall estimated $CO_2$ rates of emission from the two surfaces depicted in Figure 9 are equal to $3.42 \times 10^4$ and $1.47 \times 10^4$ ton$_{CO_2}$ year$^{-1}$ for CCR1 and CCR2, respectively. Even as the activation surface associated with CCR2 is characterized by a remarkably smaller specific rate of emission, the order of magnitude of the ensuing overall gas generation is similar to the one of CCR1. Moreover, the shape of the activation surface (in both Figures 8-9) is significantly influenced by the basin structure which may lead to discontinuities in the spatial structure of the $CO_2$ sources. The basin structure and stratigraphy

are then key factors driving the amount of $CO_2$ potentially generated by CCR mechanisms. As such, while the methodological framework we present is general, the results are case-specific and an appropriate quantification of the uncertainty associated with the geological setting is always required to constrain modeling results.

| CCR | $\mu(F_{CO_2})$ [kg $CO_2$ m$^{-2}$ Ma$^{-1}$] | $\sigma(F_{CO_2})$ [kg $CO_2$ m$^{-2}$ Ma$^{-1}$] |
|------|--------------------------------|---------------------------------|
| CCR1 | $1.797 \times 10^4$ | 16 |
| CCR2 | $1.958 \times 10^3$ | 16 |

Table 4. Mean($\mu$) and standard deviation ($\sigma$) computed for $F_{CO_2}$ computed for activation surfaces depicted in Figure 9.

## 5    Conclusions

We perform a probabilistic assessment of $CO_2$ generation by considering the effect of a variety of Carbonate/Clay Reactions

in a realistic large-scale three-dimensional sedimentary setting. Our work is grounded on the Carbonate/Clay Reaction (CCR) modeling approach first proposed by Ceriotti et al. (2017) which has been showcased in a one-dimensional set-up and embeds a framework for quantification and propagation of uncertainty associated with thermodynamic parameters driving CCRs. In summary, the methodological approach we propose and the ensuing results can contribute to enhance our understanding on the strength of the controls of diverse geochemical mechanisms on $CO_2$ dynamics in subsurface environments, with potential

implications to several fields of practical interest, including, e.g., Carbon Capture and Storage (CCS, Metz et al., 2005), large scale groundwater flow modelling Kissinger et al. (2017) and Enhanced Oil Recovery (EOR, Allis et al., 2001; Hutcheon and Abercrombie, 1990) practices.

Here, we consider a three-dimensional system with a diagenetic history feasibly encountered in a real geological setting. We analyze the impact of three different CCR formulations and mineral assemblage on ($i$) the probability of CCR activation as a





function of temperature and pressure; $(ii)$ the frequency of activation as a function of depth; and $(iii)$ the shape and extent of the surface delimiting the three-dimensional $CO_2$ source. Our study leads to the following major conclusions:

1. The temperature and pressure of activation depend on the CCR considered. Modifying the reference CCR can lead to a markedly different scenario in terms of depth of the source and extent of the activation surface. Our stochastic framework allows quantifying the (spatially- and temporally-dependent) probability distribution of the activation temperature and pressure associated with a given CCR. With reference to the depositional setting here analyzed, non-zero probabilities of $CO_2$ generation are associated with temperature and pressure equal to 159°C and 569 bar, respectively. These vaules are relatively small if compared to those typically observed in sedimentary basins and support the potential of CCR mechanisms to act as $CO_2$ source in diagenetic environments. Notably, activation of CCR in our showcase scenario might be feasible even at a depth of 3.2 km, i.e., at location compatible with the average depths of a typical gas extraction well (i.e., $\simeq 2.5$ km). This result is of particular interest because the occurrence of shallow $CO_2$ sources reduces the $CO_2$ migration path towards hydrocarbon reservoirs, thus increasing the potential risk that the $CO_2$ generated by CCR might reach the shallow cap-rock without being precipitated as newly formed carbonates, diluted or re-dissolved in water.

2. Our work suggests the need for a fully three-dimensional assessment to describe the extent and the shape of $CO_2$ generating source and the associated specific $CO_2$ generation rate. These are the two key elements contributing to the estimation of the amount of $CO_2$ generated by a given CCR mechanism. Scenarios characterized by different surface specific rates and source areas might lead to similar overall amount of $CO_2$ generated per unit of time. We document the benefits resulting from the implementation of a three-dimensional probabilistic quantification of the main features of CCR activation temperature and pressure and pose the basis for the probabilistic assessment of $CO_2$ accumulation in subsurface systems upon relying on physically-based modeling.

3. We show that the shape of the $CO_2$ generating source is closely dependent on the basin structure and stratigraphy. Thus, the overall amount of $CO_2$ generated in a sedimentary basin requires a site-specific assessment, fully embedding uncertainty quantification. In this context, our modeling approach and probabilistic framework are readily transferable to other cases of interest to design site-specific studies.

Our methodology considers a single type of uncertainty source, i.e., the system thermodynamic parameters. As a future development, one can envision exploring the effects of multiple sources of uncertainty, including model and parametric uncertainties. Key points of interest include the study of: $(i)$ the impact of qualitatively and quantitatively different mineralogical compositions and heterogeneous spatial arrangement on CCR activation and $CO_2$ generation rate; $(ii)$ the joint occurrence of CCR and other processes; $(iii)$ the contribution to CCR characteristic activation temperature and pressure of uncertainties associated with parameters and factors embedded into the burial model (e.g., burial model boundary conditions, sediment thermal and mechanical properties).

*Data availability.* Data can be accessed at https://data.mendeley.com/datasets/nmbzst46jm/draft?a=0c3d3bc7-4d9b-4a7d-911b-e5dcfdb31b86.



*Author contributions.* G.C. performed numerical simulations, contributed to design the research methodology, analyzed data, created figures, wrote the first draft; C.G. contributed to design the research methodology, contributed data; M.D.R. provided research funding, contributed data; A.G. supervised the research, contributed to design the research methodology, discussed the results, edited the text; G.P. supervised the research, contributed to design the research methodology, discussed the results, edited the text.


*Competing interests.* A.G. is member of the Editorial Board of the Journal.

*Acknowledgements.* The study is financed by the Eni SpA R&D project "Gas Systems – Basin Gas Systems Risk Analysis". Thanks are due to the Eni management for the permission to publish the present work.



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





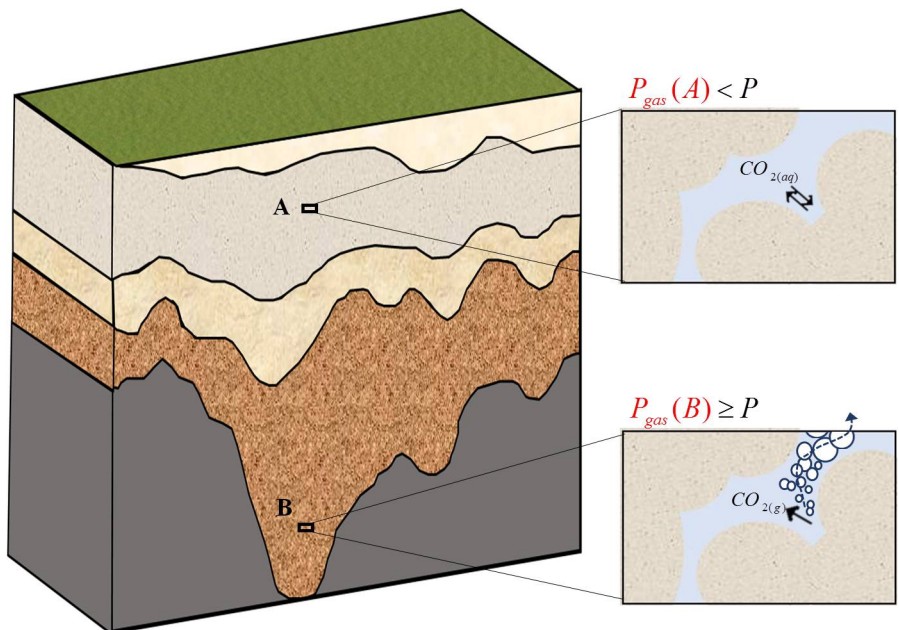

**Figure 1.** Outline of the two possible alternative scenarios according to the conceptual approach proposed in Cathles and Schoell (2007). At location A (characterized by shallow location and moderate $T$-$P$ values), the geochemical system is at equilibrium, the total gas pressure given by sum of all gases species partial pressures ($P_{gas}$) is smaller than pore-water pressure ($P$), and the $CO_2$ exists only as dissolved species. At location B (characterized by deep location where high $T$-$P$ values are expected), the total gas pressure is larger than pore-water pressure. Then a $CO_2$-rich gaseous phase is formed, which migrates upwards, and a disequilibrium is promoted leading to continuous release of $CO_2$ until one of the reactants of CCR is exhausted.





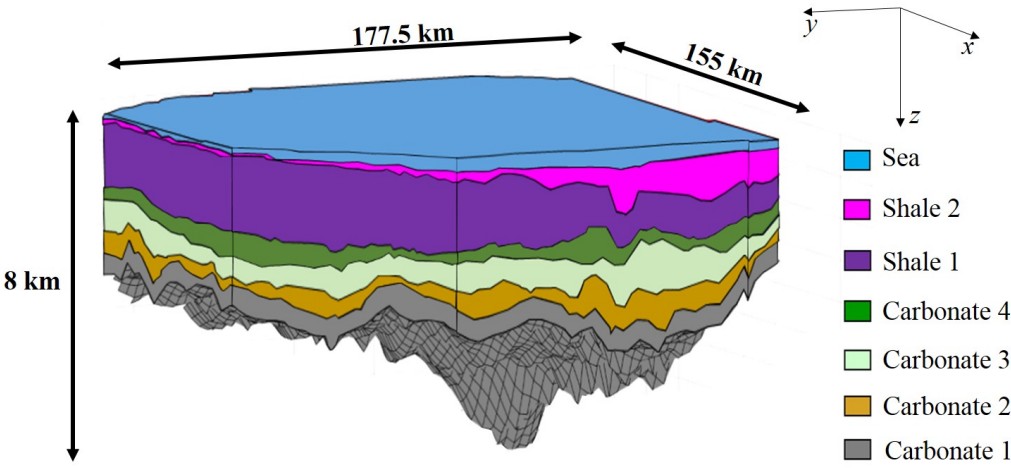

**Figure 2.** Stratigraphy of the three-dimensional realistic sedimentary basin case study at present time, i.e., $t = 0$ Ma.



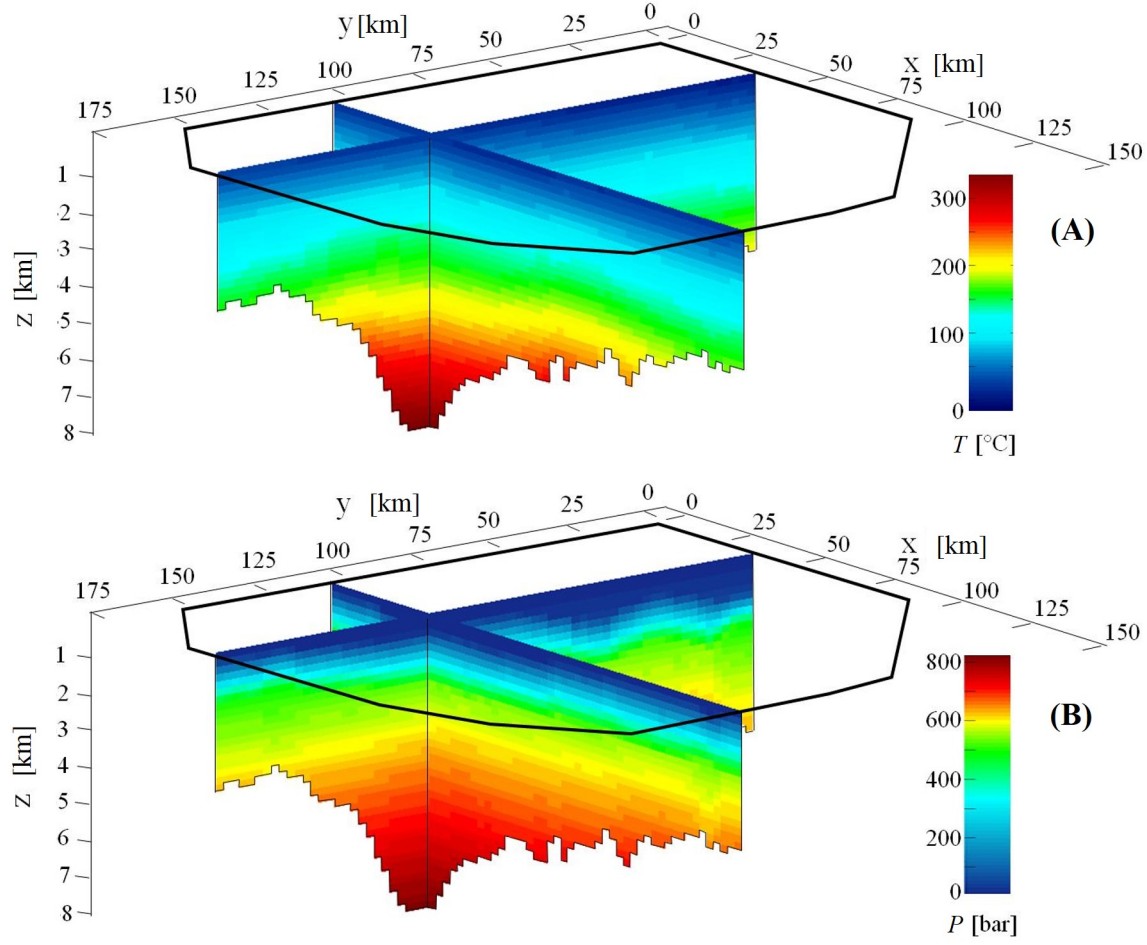

**Figure 3.** Evolution of temperature ($T$ [°C], panel A) and pressure ($P$ [bar], panel B) simulated at the present time, i.e., $t = 0$ Ma, along two perpendicular planar sections at $x = 32$ km and $y = 105$ km.





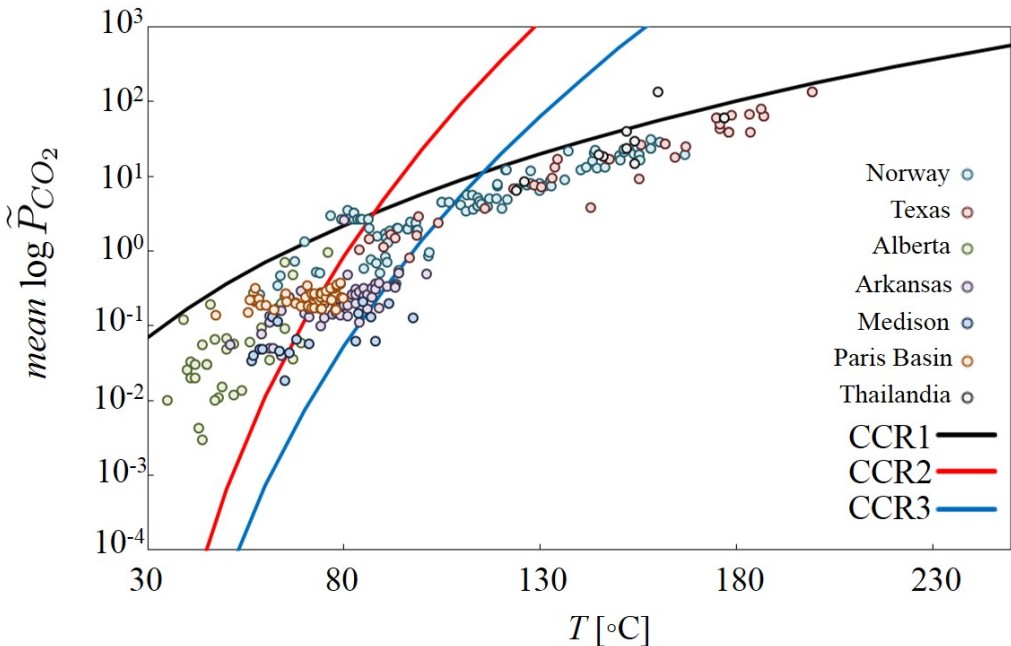

**Figure 4.** Evolution of mean log $\widetilde{P}_{CO_2}$ trend as a function of temperature computed for CCR1, CCR2 and CCR3 assuming that $P$ and $T$ are described by $P$ [bar] = 6× ($T$ [°C]-22) sugegsted by (suggested by Smith and Ehrenberg, 1989). As a term of comparison, $P_{CO_2}$ measured in different sedimentary basins labeled Norway, Texas, Alberta, Arkansas, Medison, Paris Basin, Thailandia (reported in Coudrain-Ribstein et al., 1998).





**Figure 5.** Spatial distribution of $f(\widetilde{Z}_{act})$ computed at $t = 0$ Ma along two planar perpendicular sections of the basin case study located at $x$ = 32 km and $y$ = 105 km for CCR1 (A), CCR2(B) and CCR3 (C).





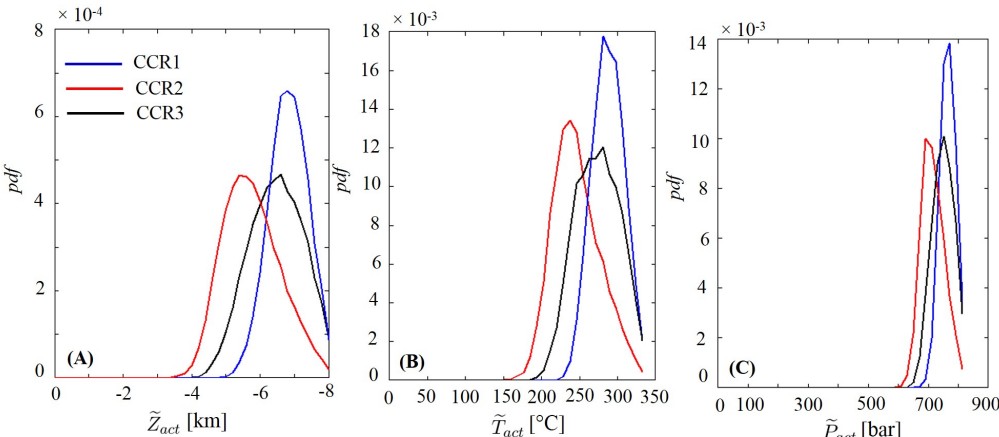

**Figure 6.** Probability density functions ($pdfs$) of $\widetilde{T}_{act}$ (A), $\widetilde{P}_{act}$ (B) and $\widetilde{Z}_{act}$ (C) computed for CCR1 (solid blue line), CCR2 (solid red line) and CCR3 (solid black line) computed at $t = 0$ Ma.





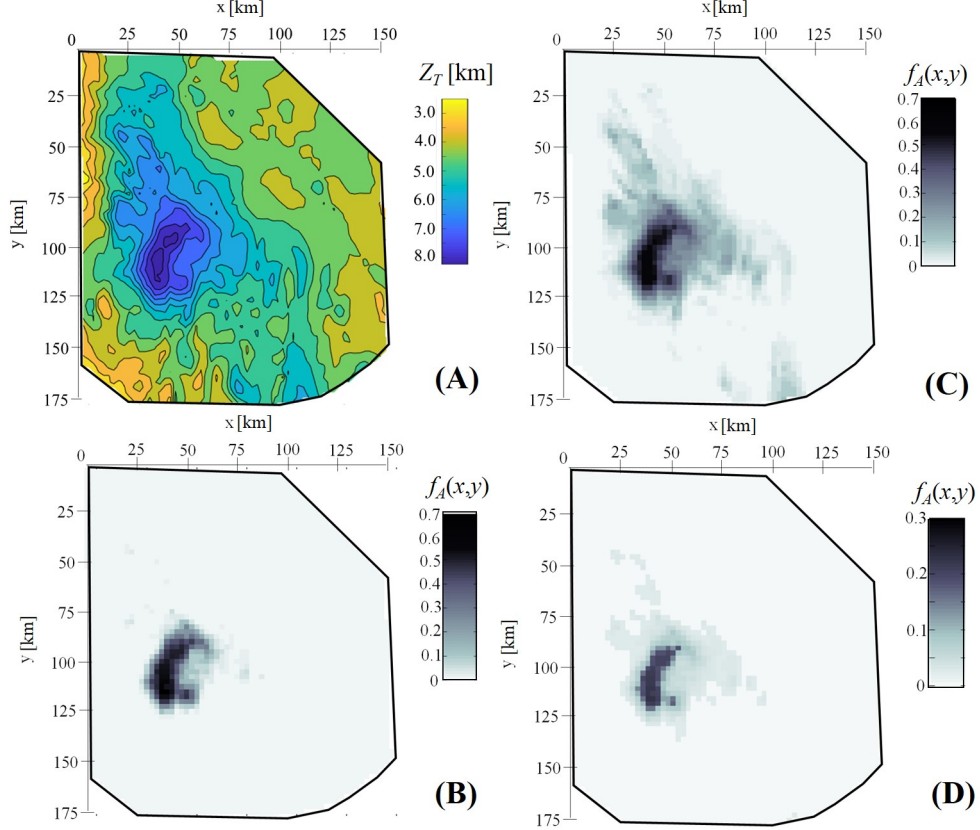

**Figure 7.** Maximum depth attained in each point of the basin $Z_T$ (A) and spatial distribution of $f_A(x,y)$, i.e. the total frequency of CCR activation for each combination of $x$ and $y$ coordinates for the corresponding column of sediments, throughout the planar surface of the basin case study at $t = 0$ Ma associated with CCR1 (B), CCR2 (C), CCR3 (D).





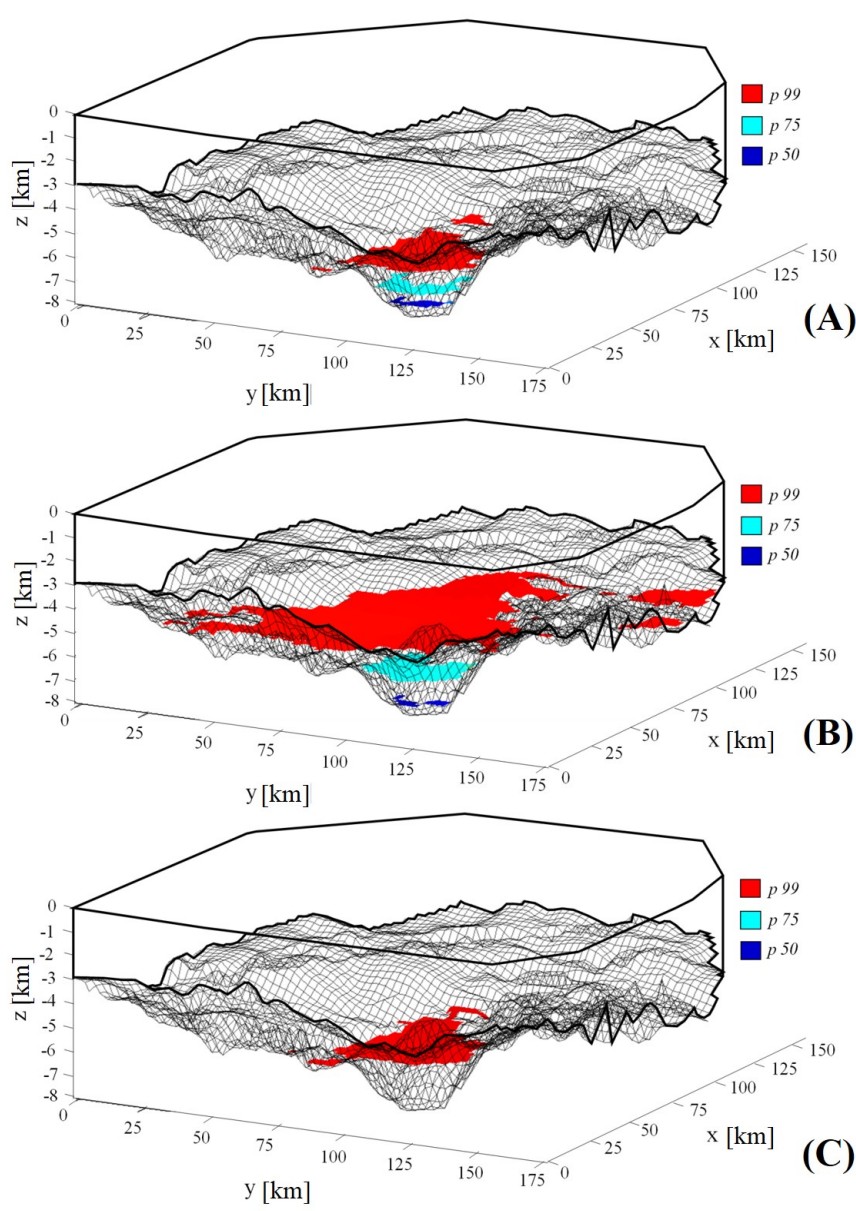

**Figure 8.** Three-dimensional illustration of activation surfaces yielded by $pp_{50}(CO_2)$ (dark blue), $pp_{75}(CO_2)$ (light blue) and $pp_{99}(CO_2)$ (red) for CCR1 (A), CCR2 (B) and CCR3 (C).

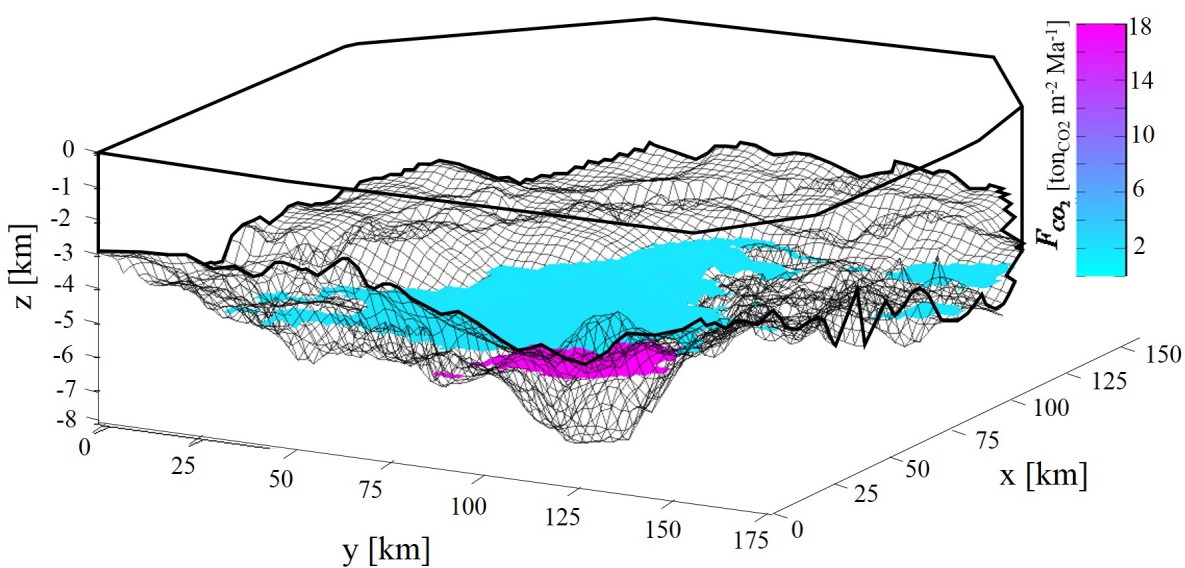

**Figure 9.** Three-dimensional illustration of activation surface yielded by $pp_{99}(CO_2)$ for CCR1 and CCR2 and the corresponding $CO_2$ generation rate for each point of the activation surfaces [kg $CO_2$ m$^{-2}$ Ma$^{-1}$].