# Peer review of "Probabilistic modeling of field-scale CO2 generation by Carbonate/Clay Reactions in sedimentary basins"

_Hydrology and Earth System Sciences, 2020_

## Referee Comment (RC1) · Anonymous Referee #1 · 1 Dec 2020

The manuscripts provides a showcase for a probabilistic framework at hand of a realistic 3D scenario. The processes under investigation generate CO2 from complex Carbonate/Clay Reactions in deep sedimentary basins where pressures are high and temperatures play a role in fundamental reactive systems. In the following and in the manuscript, these systems are denoted as CCR. As I understand the motivation for this study, it extends previous work of a lower-dimensional concept into full 3D and may be seen as a proof of concept.

We have here a manuscript that falls within the scope of HESS, presents a novel concept with well explained scientific methods. The title is not wrong, but maybe promises
too much. It is rather a proof of concept for a probabilistic framework, with still a way to go for an 'assessment'. So, I can support this study for publication in HESS, while I have some major concerns, which I am confident can be addressed.

General comments:

I) The research question(s) that guide(s) this work should be formulated more clearly, and be addressed more clearly. I see here two fields where new knowledge is produced, (i) about the probabilistic framework itself, its concept, limitations, applicability, (ii) new insights on CCR and their role in generating CO2 in sedimentary basins (e.g. the statement in Line 295/296 on pressure playing a major role in CCR activation though having minor impact on reaction equilibria). Accordingly, the structure of the manuscript, and in particular the discussion and conclusions should reflect this: in the introduction, in the results, and in the conclusions.

On the physics part, I took home messages like, e.g., pressure matters to build a gas phase, temperature affects equilibrium constants, thus burial depth; or: one CCR is less likely activated while it produces more CO2 than another one. And regarding the probabilistic framework, I am actually not sure what I learned here, see my next point.

II) The probabilistic framework includes some uncertainties (on equilibrium constants) while it takes strong assumptions in many instances. This is acknowledged in the last remarks in the Conclusions section. In fact, the list of uncertainties is endless. Understanding even more extended probabilistic frameworks becomes even more difficult, and I fear results get lost in a smoke bomb of probabilistic interpretations. That statement, of course, is exaggerated. But seriously, regarding the classification of uncertainties: first of all, I found it not easy to understand from Abstract and Introduction what the motivation for the "probabilistic assessment" is, i.e. where are the uncertainties. Examples might be given in the Introduction. Otherwise, it takes until Section 3 to find it out. Or in Lines 139-141: my impression is that deciding the mineralogical compositions are uniformly distributed is a strongly simplifying assumption, although

anything else would only increase complexity, not reliability in any sense. But does this not reduce the informative value of the overall framework, when such crucial statistical uncertainties are neglected.

Therefore: It might be helpful to put the probabilistic framework into a broader context of a clear classification of uncertainties occurring in the application of the framework. E.g. Walker et al. (2003) use definitions of different categories, like determinism, statistical uncertainty, scenario uncertainty, etc. This might help keeping an overview and in interpreting the results.

@article{walker2003, author = {Walker, Warren E and Harremo{\"e}s, Poul and Rotmans, Jan and van der Sluijs, Jeroen P and van Asselt, Marjolein BA and Janssen, Peter and Krayer von Krauss, Martin P}, journal = {{Integrated Assessment}}, number = {1}, pages = {5–17}, publisher = {Taylor \& Francis}, title = {Defining uncertainty: a conceptual basis for uncertainty management in model-based decision support}, volume = {4}, year = {2003}, }

@book{morgan1990, author = {Morgan, Millett G and Henrion, Max}, publisher = {Cambridge University Press}, title = {Uncertainty: A Guide to Dealing with Uncertainty in Quantitative Risk and Policy Analysis}, year = {1990}, }

Specific comments:

1) Line 154-155: How should I understand the statement that the equilibrium constant of speciation reactions is the key source of uncertainty? Does this mean a bigger uncertainty than spatial distribution/heterogeneity and choice of CCR? Probably not.

2) Line 213-214: Compatible sounds not wrong and I think, it is absolutely correct to use this wording. The data do not disagree with the model, but what is the message in concluding that the model does not contradict observations? Compatible sounds so weak that it demands for more explanation and interpretation. I am also not sure, but maybe the authors have an idea.

3) Conclusions: Am I right to assume that you suggest for a beneficial application of this framework, the CCR should be known in advance?

4) Conclusions, Lines 411ff: I do not really get the message of this statement. What exactly does 'physically-based modeling' refer to?

Minor comments, typos, etc.:

a) Abstract, Line 3: I was wondering if 'mono-dimensional' is a known expression. If so, I am Ok.

b) Line 25: 'relatively'

c) Lines 92, 210, and 212: 'formulations'

d) Line 162: it IS always possible

e) Line 312: is depicted (blank missing)

f) Line 401: 'values'

—————————————————

---

## Referee Comment (RC2) · Anonymous Referee #2 · 19 Jan 2021

This paper outlines a probabilistic framework to assess the evolution of CO2 in a "realistic" 3-D deep sedimentary basin. It builds on a relatively recent study (in GCA) also led first author G. Ceriotti that explored a method to quantify the impact of one type of carbonate/clay reaction (CCR) in a 1-D basin scenario. In this follow-on study they expand their assessment to include other types of CCFR. The overarching goal of this new study is to interrogate, in 3-D, the distribution of CO2 generated by the CCRs in a temperature-pressure regime dictated by the boundary conditions of their basin model system.

Scientific significance: Does the manuscript represent a substantial contribution to scientific progress within the scope of Hydrology and Earth System Sciences (substantial new concepts, ideas, methods, or data)?

Understanding the behavior of C-O-H fluids in sedimentary basins is certainly a timely topic especially with the recent emphasis on the extraction of gas and oil from tight formations and the potential for storage of $CO_2$ in the subsurface to mitigate anthropogenic greenhouse gas emissions. This paper tackles this problem by defining a 3-D basin burial scenario dominated mainly by dolomitic rock units with overlying shale caprock. They refer to a widely tested and documented burial model known as E-SIMBA which admittedly I have not heard of before (TOUGH, TOUGH2, iTOUGH and TOUGHREACT are examples of codes familiar to this reviewer). If their burial model tied to a probabilistic approach tracks fluid evolution via the carbonate-centric reactions they identify while at the same time documenting the evolution of porosity and permeability leading to the $CO_2$ distributions they identify, then perhaps yes this may be a nice contribution to the understanding of basin processes. However, based on what has been presented, I found it somewhat difficult to assess how their outcomes connected these coupled processes. I would like to have seen how porosity and permeability evolve during the burial process and how in turn these are related to the true distribution of $CO_2$. Their visualizations illustrate where $CO_2$ is enriched but they seem to cover a wide region of certain horizons. I guess this just means there is a high probably of finding a specific $FCO_2$ value in this area.

Rating: Good

Scientific quality: Are the scientific approach and applied methods valid? Are the results discussed in an appropriate and balanced way (consideration of related work, including appropriate references)?

The application of probabilistic methods with associated statistical underpinning and Monte Carlo simulation is certainly one way to assess geochemical processes in an evolving sedimentary basin. The goal is to track the three reaction types in space and

time as the basin evolves. The frequency of CCR and associated distribution of the resulting $CO_2$ reaction product are visually represented in 3-D very clearly (Fig. 7-9). I am very appreciative of their attempt to constrain the different levels of uncertainty but must admit I found their narrative describing uncertainly somewhat unwieldy and difficult to follow. Further this made it hard for me to understand how these uncertainties influenced the kinds of outcomes they represented on the key Figures 6 – 9. Perhaps it is not fair to criticize the nature of how they defined their sedimentary system, but I do have to wonder about the justification for selecting a dolomitic-rich rock as one of the starting lithologies. Dolostones are certainly not uncommon in the sedimentary record, reportedly making up to 2 percent of crustal rocks. However, a large percentage of the dolomite in thick marine dolostone units is thought by many geologists and geochemists to have been formed by replacement of $CaCO_3$ sediment rather than by direct precipitation. This authigenic process can start near the surface but is certainly facilitated by deeper burial involving the evolution and transport of Mg-rich brines infiltrating the calcite-rich formation; this reaction can yield a pretty big increase in porosity up to 14%. So, to me a more "realistic" basin scenario would be to start with a limestone, alter it to dolomite during burial with the associated porosity (and permeability) change, and then with deeper burial initiate the alteration reactions of the sort they identify. I also appreciate the impetus for picking specific simple mineral assemblages as a starting point for the modeling, but beidellite is not a phase typically observed in deep shale systems. And I have yet come across a shale (mudstone) with 42% microcline; this level of feldspar plus 50% clay would make this a very unusual rock. There are few things I am concerned about with respect to the reactions they picked. These represent just a very small number of possible reactions that could take place during burial. So why not define the starting mineralogy, initiate the burial process of increasing P and T, and let thermodynamics drive the water-rock interactions to the most favored stable reactions. A priori selection of the reactions seems rather narrow in thinking, although I do appreciate, they wanted to target the most optimum reactions to produce $CO_2$ but is this truly "realistic". Second, they consider a system where

the fluid is pure water which is very unrealistic when is it comes to sedimentary basin fluids; most are saline (typically 50-100 g/kg TDS). This would change the activity of water and in turn impact the solubility of the $CO_2$ (the salting-out effect). High concentrations of $CO_2$ would also affect the activity water. And to be sure a different activity of water would impact the nature of the reactions they did identify. Third, they seem to pull thermodynamic data from what I consider outdated references. For example, Ian Hutcheon's work is certainly respected, but the authors should be very careful using thermodynamic data/insights that date back over 20 years. I recommend the authors take a journey through some the sources provided here (and associated references) just to be sure they are on the right path (this falls into the category of capturing "thermodynamic uncertainty"):

Modeling Metamorphic Rocks Using Equilibrium Thermodynamics and Internally Consistent Databases: Past Achievements, Problems and Perspectives Pierre Lanari, Erik Duesterhoeft Journal of Petrology, Volume 60, Issue 1, January 2019, p. 19-56 https://doi.org/10.1093/petrology/egy105

CHNOSZ: Thermodynamic Calculations and Diagrams for Geochemistry Jeffrey M. Dick Front. Earth Sci., 16 July 2019 https://doi.org/10.3389/feart.2019.00180

Thermodynamic Data for Geochemical Modeling of Carbonate Reactions Associated with $CO_2$ Sequestration – Literature Review (only focuses on carbonates but still may be useful) KM Krupka KJ Cantrell BP McGrail: September 2010 Prepared for the U.S. Department of Energy under Contract DE-AC05-76RL01830

Qualification of Thermodynamic Data for Geochemical Modeling of Mineral-Water Interactions in Dilute Systems T. J. Wolery and C. Jove-Colon ANL-WIS-GS-000003 REV 00 November 2004

Zimmer, K., Zhang, Y.L., Lu, P., Chen, Y.Y., Zhang, G.R., Dalkilic, M. and Zhu, C. (2016) SUPCRTBL: A revised and extended thermodynamic dataset and software package of SUPCRT92. Computer and Geosciences 90:97-111.

doi.org/10.1016/j.cageo.2016.02.013

Rating: Good (leaning toward Fair)

Presentation quality: Are the scientific results and conclusions presented in a clear, concise, and well-structured way (number and quality of figures/tables, appropriate use of English language)?

Given the approach they adopted, the results and conclusions seem reasonable. But that said, this does not make them necessarily significant or innovative. They took an interesting approach with their probabilistic modeling then seemed to engineer the outcome they wanted by orchestrating a narrow type of reaction chemistry rather than truly allowing a more realistic mineralogic system to evolve during burial. I appreciate the probabilistic approach differs significantly from say a full developed 3-D reactive transport model that allows the system to track the thermodynamically most favorable reactions with accompanying fluid evolution. And this of course only considers the system from an equilibrium thermodynamic point of view. Although more difficult, one could also address the evolution of the model system by quantitively assessing where and when the system deviates from equilibrium during burial. I guess I was hoping for better articulation of the connection between the evolution of the mineralogy and $CO_2$ and the changes in porosity-permeability. For example, was $CO_2$-rich fluid allowed to migrate from one unit to another or was every rock unit treated as a closed system. The probabilistic approach is certainly interesting but seems to fail in capturing the dynamics of a complex heterogenous system undergoing change over non-trivial length and time scales. So, what did we learn from this paper? We learned that if you take a dolomite-bearing or dolomite-rich rock containing other phases like some clay or feldspar and push the rock to higher P and T approaching low-grade metamorphism in a deep basin you can make lots of $CO_2$. I think we kind of already knew this. What would have been really interesting to see is how this $CO_2$ concentration changed as burial proceeded for each rock unit as a function of space and time, and how these changes affected the porosity and permeability of each unit. I am thinking they may

have such information and if so, I encourage them to expand their outcomes to be more inclusive.

Rating: Good
* * *

---

## Referee Comment (RC3) · Anonymous Referee #3 · 26 Jan 2021

This manuscript provides a three-dimensional test case study to assess the generation of CO2 by carbonate/clay reactions in a statistical framework. The authors extend the work of Ceriotti et al (2017) to a synthetic field setting. The article provides nice insights about the generation and source location of CO2. I think that the article deserves publication after some minor corrections that should address several points: (1) the authors should take some time to explain the source of uncertainty in equilibrium constants. Explain why the coefficients associated with equilibrium constants are random and which is the underlying process. Also, it is important to discuss the parameters describing this uncertainty from a realistic point of view. Are these values realistic? Do they compare with a real field? It is not clear why the authors state that equilibrium constant

[Figure]

Ks,ph depends on pressure and temperature but equation (1) only has temperature and is valid only for p=1 bar; (2) some of the assumptions are not clear but from the introduction I was expecting some two-phase flow simulations besides generation; (3) the authors state too many times that something can be found in Ceretti et al (2017), this makes the description of methods not self-contained; (4) not clear what do you mean by phases in equation (2) and reaction CCR, may be this is too abstract, and the authors should give an example to follow; (5= sorry if I miss this one but I do not see the definition of C_A, not clear how is it calculated. Other than this, I think that the manuscript is well written and organized and will be a nice contribution for HESS. I did not see typos except in line 162 where you are missing "it is".

---

## Author Comment (AC1) · 14 Mar 2021

Dear Reviewer:

We appreciate the efforts you have invested in reviewing our manuscript. We are now providing our responses to the comments received for your consideration. In the

following reviewer comments are in italic, our responses are in plain text, proposed changes in blue.

Sincerely,

Giulia Ceriotti (on behalf of all the authors)

*The manuscript provides a showcase for a probabilistic framework at hand of a realistic 3D scenario. The processes under investigation generate CO2 from complex Carbonate/Clay Reactions in deep sedimentary basins where pressures are high and temperatures play a role in fundamental reactive systems. In the following and in the manuscript, these systems are denoted as CCR. As I understand the motivation for this study, it extends previous work of a lower-dimensional concept into full 3D and may be seen as a proof of concept. We have here a manuscript that falls within the scope of HESS, presents a novel concept with well explained scientific methods. The title is not wrong, but maybe promises too much. It is rather a proof of concept for a probabilistic framework, with still a way to go for an 'assessment'. So, I can support this study for publication in HESS, while I have some major concerns, which I am confident can be addressed.*

We thank the Reviewer for their analysis of our work. We understand that the wording "assessment" in the title might have caused some misunderstanding on the key objective of our study. Thus, in our revisions we have taken also this element into account and have revised the manuscript Title and Abstract to sharpen the focus of our study.

Title is modified as follows:

"Probabilistic modeling of field-scale CO2 generation by Carbonate/Clay Reactions in sedimentary basins".

Accordingly, we rephrase the Abstract at lines 1-2 as follows
"Abstract. This work explores a probabilistic modeling workflow and its implementation targeting CO2 generation rate and CO2 source location by occurrence of Carbonate/Clay Reactions (CCRs) in three-dimensional realistic sedimentary basins. We ground our study on the [...]".

To avoid any misunderstanding, we recast some sentences in Sections 4 and 5 as follows:

- At line 247 "We tackle probabilistic modeling of the CCRs introduced in Section 1 upon relying on a numerical Monte Carlo (MC) approach".

- At line 300 "According to our probabilistic modeling framework, the distribution of [...]".

- At line 326 "Our probabilistic model documents that the characteristic temperature and pressure [...]".

- At line 384 "We rely on a probabilistic modeling framework to model CO2 generation [...]".

General comments:
*I) The research question(s) that guide(s) this work should be formulated more clearly, and be addressed more clearly. I see here two fields where new knowledge is produced, (i) about the probabilistic framework itself, its concept, limitations, applicability, (ii) new insights on CCR and their role in generating CO2 in sedimentary basins (e.g. the statement in Line 295/296 on pressure playing a major role in CCR activation though having minor impact on reaction equilibria). Accordingly, the structure of the*

*manuscript, and in particular the discussion and conclusions should reflect this: in the introduction, in the results, and in the conclusions. On the physics part, I took home messages like, e.g., pressure matters to build a gas phase, temperature affects equilibrium constants, thus burial depth; or: one CCR is less likely activated while it produces more CO2 than another one. And regarding the probabilistic framework, I am actually not sure what I learned here, see my next point.*

We consider uncertain equilibrium thermodynamic constants because sedimentary settings require considering temperature and pressure conditions lying outside the ranges where these parameters are usually appraised. A similar choice was illustrated in one of our previous studies (Ceriotti et al., 2017). Here, we show how considering this source of parametric uncertainty propagates to key outputs when a three-dimensional scenario and alternative mechanisms are considered, thus incorporating an additional source of uncertainty. Note that considering a three-dimensional case is relevant because it allows obtaining a space-resolved delineation of the identified CO2 sources and a quantification of the related fluxes.

While we prefer not to change significantly the structure of the manuscript, we have revised the Introduction and our Conclusions to pinpoint the key elements of innovation associated with the probabilistic framework illustrated in this work.

The revised Introduction now reads (starting from line 93 of the original manuscript):
"Modeling of CO2 generation and accumulation in large-scale geological systems is typically prone to considerable uncertainties, chiefly due to paucity of information and to the large spatial and temporal scales involved. In this context, we provide a modeling framework that leads to a probabilistic quantification of the generation of CO2 by a specific class of reactive processes (i.e., CCRs). As such, our study fills a knowledge gap by providing a methodology to support quantitative investigations of spontaneous CO2 generation in large scale geological systems, these being otherwise typically based on mostly qualitative analyses. While we consider a simple geochemical model based on thermodynamic equilibrium, our probabilistic framework of analysis is flexible and can

include treatment of model uncertainty (Walker, 2003; Neuman, 2003) as an additional element. Setting a given model structure is simply a convenient choice to minimize computational and conceptual complexity while at the same time considering a mathematical model that can be characterized with information that is typically available in field scale settings (in terms of, e.g., mineral composition, pressure, and temperature distributions). Values of equilibrium constants are here considered as uncertain because temperature and pressure values observed in sedimentary systems lie outside the range of conditions where these parameters are usually characterized (Ceriotti et al., 2017; Blanc, 2012). In this work we investigate the propagation of this parametric uncertainty in the presence of various (alternative) CCR formulations by focusing on a three-dimensional scenario. When considering the framework proposed by Walker et al. (2003), our work allows combining uncertainty in model parameters (equilibrium thermodynamic constants) with input uncertainty, i.e., uncertainty in the description of the reference system. The latter type of uncertainty is reflected by our choice of considering diverse mineral assemblages leading to the occurrence of different CCRs. Note that our approach is geared towards quantification on the space-time location and intensity of the $CO_2$ source. This information can then be used as input to quantify scenario uncertainties, by delineating the spatial and temporal extent of $CO_2$ influx. Transport and accumulation of $CO_2$ across the subsurface can then be analyzed through approaches such as those described, e.g., in Battistelli et al. (2016). From an operational standpoint, our approach could be applied to enhance our knowledge on the degree of compatibility of $CO_2$ concentrations observed in field scale systems with the occurrence of CCR, as opposed to the action of other processes which might be considered in a large scale transport model of choice. The study is structured as follows (. . .)"

Conclusions, line 385 is modified as follows:
"Our work is grounded on the probabilistic approach proposed by Ceriotti et al. (2017) to treat Carbonate/Clay Reactions (CCR). Such an approach embeds quantification of parametric uncertainty associated with the thermodynamic equilibrium constants

driving CCR and has been showcased by these authors in a one-dimensional set-up".

Conclusions item 1 is modified as follows:

"We rely on geochemical equilibrium and quantify uncertainty associated with model parameters and inputs, the latter source of uncertainty corresponding to the uncertainty in the information required to describe the reference system (i.e., input uncertainty; Walker et al., 2003). The presence of input uncertainties implies the possibility that diverse CCRs may occur and lead to differing degrees of importance of parametric uncertainty on CO2 generation. Our stochastic framework (. . .)"

Conclusions item 2 is modified as follows: "We quantify the way the considered input and parametric uncertainty propagates onto estimates of generated mass of CO2 in a three-dimensional system. This allows describing the extent and the shape of the CO2 generating source together with the associated CO2 generation rate. (. . .)"

II)The probabilistic framework includes some uncertainties (on equilibrium constants) while it takes strong assumptions in many instances. This is acknowledged in the last remarks in the Concluions section. In fact, the list of uncertainties is endless. Understanding even more extended probabilistic frameworks becomes even more difficult, and I fear results get lost in a smoke bomb of probabilistic interpretations. That statement, of course, is exaggerated. But seriously, regarding the classification of uncertainties: first of all, I found it not easy to understand from Abstract and Introduction what the motivation for the "probabilistic assessment" is, i.e. where are the uncertainties. Examples might be given in the Introduction. Otherwise, it takes until Section 3 to find it out. Or in Lines 139-141: my impression is that deciding the mineralogical compositions are uniformly distributed is a strongly simplifying assumption, although anything else would only increase complexity, not reliability in any sense. But does this not reduce the informative value of the overall framework, when such crucial statistical uncertainties are neglected. Therefore: It might be helpful to put the probabilistic framework into a broader context of a clear classification of uncertainties occurring in the application of the framework. E.g. Walker et al. (2003) use definitions of differ-

*ent categories, like determinism, statistical uncertainty, scenario uncertainty, etc. This
might help keeping an overview and in interpreting the results.*

We thank the Reviewer for this comment which prompts us to improve the way the key
objectives of the study are framed. We also elaborate further on our choice to consider
a spatial uniform mineral assemblage.

These points are now addressed in Section 3. The introductory paragraph of this Section now reads (line 154 of the original draft):
"Our study relies on a given model structure, thus neglecting uncertainty in the latter.
We rest on the equilibrium-based approach used by Ceriotti et al. (2017). Thus, we
consider pure mineral phases while neglecting other factors which would eventually
influence the model structure (e.g., the occurrence of other mineral transformations, or
effects associated with salinity of brine). Consistent with this model structure, we consider the equilibrium constant of speciation reactions as the key source of parametric
uncertainty. We note that this choice is motivated by the observation that temperature
and pressure values observed in sedimentary systems lie outside the range of conditions where thermodynamic equilibrium constants are usually characterized (Blanc,
2012). In addition to parametric uncertainty, we also consider input uncertainty, defined
as the uncertainty related to the description of the system (Walker, 2003), i.e., we assume that diverse CCRs may take place depending on the mineralogical assemblage.
These two sources of uncertainty are propagated throughout the final modeling goals
of interest, i.e., the $CO_2$ source location, the $CO_2$ generation rate, and the temperature and pressure of CCR activation. Note that, as detailed in Section 2, we consider
a uniform mineral composition across the domain, a setting corresponding to an upper limit condition for each of the considered CCRs. While it would be interesting in
principle to investigate the impact of a spatially heterogeneous mineralogic composition, doing so would require having at our disposal on a suitable dataset and would
increase complexity. Yet, it is worth emphasizing that the proposed methodological
framework and modeling approach are fully compatible with the presence of a spatially

variable mineralogical composition, which can be accommodated in the presence of appropriate data to characterize it. As such, our approach can be employed to assess the impact of uncertainties associated with spatially heterogeneous arrangements of mineral and sediment composition on CCR-based $CO_2$ generation. The latter could be tackled upon relying on appropriate techniques such as, e.g., Functional Compositional Kriging (see, e.g., Menafoglio et al., 2016, and references therein). Analyzing this aspect is, however, beyond the scope of the present study."

Specific comments:
*1) Line 154-155: How should I understand the statement that the equilibrium constant of speciation reactions is the key source of uncertainty? Does this mean a bigger uncertainty than spatial distribution/heterogeneity and choice of CCR? Probably not.*

We are confident that the motivation at the basis of our choice is now clearly stated and has been addressed through the revisions outlined above.

*2) Line 213-214: Compatible sounds not wrong and I think, it is absolutely correct to use this wording. The data do not disagree with the model, but what is the message in concluding that the model does not contradict observations? Compatible sounds so weak that it demands for more explanation and interpretation. I am also not sure, but maybe the authors have an idea.*

In our opinion "compatible" is an appropriate expression. Identifying the source of $CO_2$ accumulations for each of the listed basins is beyond our scope. Modeling can refine and enhance our understanding of the effects of alternative processes that may contribute (jointly or exclusively) to $CO_2$ accumulations. However, only after a dedicated analysis of a specific study it would be possible to formulate hypotheses on the relevance of CCR in a real case also in comparison with other processes (e.g., magma degassing, biologically driven processes or other geochemical/geological processes). For this reason, we would prefer maintaining the wording compatible in the revised manuscript.

*3) Conclusions: Am I right to assume that you suggest for a beneficial application of*

*this framework, the CCR should be known in advance?*

When detailed information / observation on mineral compositions are available, one or more CCRs can be identified as possible $CO_2$ sources. Our framework enables one to jointly consider alternative CCRs when such information is not available, thus fully embedding the effects of such uncertainty in the desired modeling goals. This is clarified in the revised manuscript Conclusions starting from line 397.

"[. . .] Our stochastic framework allows quantifying the (spatially- and temporally-dependent) probability distribution of the activation temperature and pressure associated

*4) Conclusions, Lines 411ff: I do not really get the message of this statement. What exactly does 'physically-based modeling' refer to?*

Our revised text now reads: "(. . .) provides a proof-of-concept of the applicability of process-based probabilistic frameworks for quantitative modeling of $CO_2$ accumulation in subsurface systems".

Minor comments, typos, etc.:
*a) Abstract, Line 3: I was wondering if 'mono-dimensional' is a known expression. If so, I am Ok.*

We changed it to one-dimensional.
*b) Line 25: 'relatively'*
*c) Lines 92, 210, and 212: 'formulations'*
*d) Line 162: it IS always possible*
*e) Line 312: is depicted (blank missing)*
*f) Line 401: 'values'*

These typos are now fixed.

**1 References**

Battistelli A., Berry P., Bondua' S., Bortolotti V., Consonni A., Cormio C., Geloni C. and Vasini E. M. (2016) Thermodynamics-related processes during the migration of acid gases and methane in deep sedimentary formations. Greenhouse Gases, 7:295–312.

Blanc P., Lassin A., Piantone P., Azaroual M., Jacquemet N., Fabbri A. and Gaucher E. C. (2012) Thermoddem: a geochemical database focused on low temperature water/rock interactions and waste materials. Applied Geochemistry 27(10), 2107-2116.

Blanc P., Vieillard P., Gailhanou H., Gaboreau S., Gaucher E′., Fialips C. I., Made′ B. and Giffaut E. (2015) A generalized model for predicting the thermodynamic properties of clay minerals. Am. J. S. 315(8), 734-780.

Ceriotti G., Porta G., Geloni C., Dalla Rosa M., and Guadagnini A. (2017) Quantification of CO2 generation in sedimentary basins through carbonate/clays reactions with uncertain thermodynamic parameters, Geochimica et Cosmochimica Acta, 213, 198-215.

Menafoglio, A., Guadagnini, A., and Secchi, P.: Stochastic simulation of soil particle-size curves in heterogeneous aquifer systems through a Bayes space approach, Water Resources Research, 52, 5708-5726, 2016.

Neuman S. P. (2003) Maximum likelihood Bayesian averaging of uncertain model predictions. Stochastic Environmental Research and Risk Assessment. 17, 291-305.

Parkhurst D. L. and Appelo C. (2013) Description of input and examples for PHREEQC version 3 - a computer program for speciation, batch-reaction, one-dimensional transport, and inverse geochemical calculations. US Geological Survey Techniques and Methods, Book 6, 497.

Walker W.E , Harremoes P , Rotmans J. , Van Der Sluijs J.P. , Van Asselt M.B.A. , Janssen P. And Krayer Von Krauss M.P. (2003) Defining Uncertainty A Conceptual Basis for Uncertainty Management in Model-Based Decision Support, Integrated Assessment, 4(1), 5 - 17.

---

## Author Comment (AC2) · 14 Mar 2021

Dear Reviewer:

We appreciate the efforts you have invested in reviewing our manuscript. We are now providing our responses to the comments received for your

consideration. In the following reviewer comments are in italic, our responses are in plain text, proposed changes in blue.

Sincerely,

Giulia Ceriotti (on behalf of all the authors)

*This paper outlines a probabilistic framework to assess the evolution of CO2 in a "realistic" 3-D deep sedimentary basin. It builds on a relatively recent study (in GCA) also led first author G. Ceriotti that explored a method to quantify the impact of one type of carbonate/clay reaction (CCR) in a 1-D basin scenario. In this follow-on study they expand their assessment to include other types of CCFR. The overarching goal of this new study is to interrogate, in 3-D, the distribution of CO2 generated by the CCRs in a temperature-pressure regime dictated by the boundary conditions of their basin model system.*
*Scientific significance: Does the manuscript represent a substantial contribution to scientific progress within the scope of Hydrology and Earth System Sciences (substantial new concepts, ideas, methods, or data)?*
*Understanding the behavior of C-O-H fluids in sedimentary basins is certainly a timely topic especially with the recent emphasis on the extraction of gas and oil from tight formations and the potential for storage of CO2 in the subsurface to mitigate anthropogenic greenhouse gas emissions. This paper tackles this problem by defining a 3-D basin burial scenario dominated mainly by dolomitic rock units with overlying shale caprock. They refer to a widely tested and documented burial model known as ESIMBA which admittedly I have not heard of before (TOUGH, TOUGH2, iTOUGH and TOUGHREACT are examples of codes familiar to this reviewer). If their burial model tied to a probabilistic*

*approach tracks fluid evolution via the carbonate-centric reactions they identify while at the same time documenting the evolution of porosity and permeability leading to the CO2 distributions they identify, then perhaps yes this may be a nice contribution to the understanding of basin processes. However, based on what has been presented, I found it somewhat difficult to assess how their outcomes connected these coupled processes. I would like to have seen how porosity and permeability evolve during the burial process and how in turn these are related to the true distribution of CO2. Their visualizations illustrate where CO2 is enriched but they seem to cover a wide region of certain horizons. I guess this just means there is a high probably of finding a specific FCO2 value in this area. Rating: Good*

The codes referenced by the Reviewer are excellent examples of reactive transport computational tools aimed at modeling flow, transport, and reactive processes in subsurface environments. Burial models, such as ESIMBA used in this work, are developed to simulate geological processes governing the evolution of deep subsurface environments involving extremely slow phenomena (whose reaction rate is hard to evaluate) developing across considerably large spatial and temporal scale, such as, e.g., diagenetic processes. Evolution of porosity along with the diagenetic process is included in the model ESIMBA, as explicitly stated at line 115 in the original version of the manuscript. We did not include a figure portraying a three-dimensional spatial representation of porosity distribution (of the kind, e.g., similar to what has been done for temperature and pressure in Figure 3) because porosity variations have only a mild influence on the outputs of our study. We then point out that our approach allows characterizing CO2 sources and may be then coupled with reactive transport models such as those mentioned by the Reviewer (see also Battistelli et al., 2016). We emphasize that modeling of CO2 migration and the ensuing dynamics is beyond the scope of our study, which targets a quantitative probabilistic characterization of space-time location and intensity (quantified as source or influx, FCO2) of CO2 sources. To avoid ambiguities (for which we apologize), this is now explicitly stated in the following revised paragraph in the Introduction:

"Note that our approach is geared towards quantification on the space-time location and intensity of the CO2 source. This information can then be used as input to quantify scenario uncertainties, by delineating the spatial and temporal extent of CO2 influx. Transport and accumulation of CO2 across the subsurface can then be analyzed through approaches such as those described, e.g., in Battistelli et al. (2016). From an operational standpoint, our approach could be applied to enhance our knowledge on the degree of compatibility of CO2 concentrations observed in field scale systems with the occurrence of CCR, as opposed to the action of other processes which might be considered in a large scale transport model of choice.".

*Scientific quality: Are the scientific approach and applied methods valid? Are the results discussed in an appropriate and balanced way (consideration of related work, including appropriate references)?*

*The application of probabilistic methods with associated statistical underpinning and Monte Carlo simulation is certainly one way to assess geochemical processes in an evolving sedimentary basin. The goal is to track the three reaction types in space and time as the basin evolves. The frequency of CCR and associated distribution of the resulting CO2 reaction product are visually represented in 3-D very clearly (Fig. 7-9). I am very appreciative of their attempt to constrain the different levels of uncertainty but must admit I found their narrative describing uncertainly somewhat unwieldy and difficult to follow. Further this made it hard for me to understand how these uncertainties influenced the kinds of outcomes they represented on the key Figures 6 – 9.*

We thank the Reviewer for appreciating the approach and the figures that we have proposed.

We do hope that the revised manuscript can facilitate appraising the main focus of our study, which is keyed to the formulation of a flexible stochastic modeling framework capable of embedding diverse sources of uncertainties in the target environmentally relevant scenario. This is now further elaborated in Section 3, where we add the following revised text:

"Our study relies on a given model structure, thus neglecting uncertainty in the latter. We rest on the equilibrium-based approach used by Ceriotti et al. (2017). Thus, we consider pure mineral phases while neglecting other factors which would eventually influence the model structure (e.g., the occurrence of other mineral transformations, or effects associated with salinity of brine). Consistent with this model structure, we consider the equilibrium constant of speciation reactions as the key source of parametric uncertainty. We note that this choice is motivated by the observation that temperature and pressure values observed in sedimentary systems lie outside the range of conditions where thermodynamic equilibrium constants are usually characterized (Blanc, 2012). In addition to parametric uncertainty, we also consider input uncertainty, defined as the uncertainty related to the description of the system (Walker, 2003), i.e., we assume that diverse CCRs may take place depending on the mineralogical assemblage. These two sources of uncertainty are propagated throughout the final modeling goals of interest, i.e., the $CO_2$ source location, the $CO_2$ generation rate, and the temperature and pressure of CCR activation. Note that, as detailed in Section 2, we consider a uniform mineral composition across the domain, a setting corresponding to an upper limit condition for each of the considered CCRs. While it would be interesting in principle to investigate the impact of a spatially heterogeneous mineralogic composition, doing so would require having at our disposal on a suitable dataset and would increase complexity.

Yet, it is worth emphasizing that the proposed methodological framework and modeling approach are fully compatible with the presence of a spatially variable mineralogical composition, which can be accommodated in the presence of appropriate data to characterize it. As such, our approach can be employed to assess the impact of uncertainties associated with spatially heterogeneous arrangements of mineral and sediment composition on CCR-based CO2 generation. The latter could be tackled upon relying on appropriate techniques such as, e.g., Functional Compositional Kriging (see, e.g., Menafoglio et al., 2016, and references therein). Analyzing this aspect is, however, beyond the scope of the present study."

*Perhaps it is not fair to criticize the nature of how they defined their sedimentary system, but I do have to wonder about the justification for selecting a dolomitic-rich rock as one of the starting lithologies. Dolostones are certainly not uncommon in the sedimentary record, reportedly making up to 2 percent of crustal rocks. However, a large percentage of the dolomite in thick marine dolostone units is thought by many geologists and geochemists to have been formed by replacement of CaCO3 sediment rather than by direct precipitation. This authigenic process can start near the surface but is certainly facilitated by deeper burial involving the evolution and transport of Mg-rich brines infiltrating the calcite-rich formation; this reaction can yield a pretty big increase in porosity up to 14%. So, to me a more "realistic" basin scenario would be to start with a limestone, alter it to dolomite during burial with the associated porosity (and permeability) change, and then with deeper burial initiate the alteration reactions of the sort they identify.*

We thank the Reviewer for pointing out an alternative interesting scenario of sedimentary basin evolution. The probabilistic approach presented in this work is fully compatible with the sedimentary basin scenario suggested by the Reviewer. The latter setting can be included in our workflow by replacing the

spatially homogeneous mineralogy scenario considered in the manuscript with a scenario according to which mineralogy varies with the temporal progressing of the burial process, eventually capturing the transformation of calcite into dolomite.

We clarify that the terminology realistic we use referring to the basin considered is linked to its geometry and pressure and temperature ranges and distribution. As stated in the original manuscript (lines 136-138), the mineralogical assemblage is selected to maximize the generation of $CO_2$ based on the stoichiometric balance of the CCRs analyzed. The purpose of presenting such simple mineralogical composition is to focus on (a) demonstrating the applicability of the proposed probabilistic workflow to three-dimensional systems and (b) illustrating how to analyze and interpret the richness of information that can be obtained employing such a methodology when investigating a case study of interest given a selected mineralogical assemblage.

*I also appreciate the impetus for picking specific simple mineral assemblages as a starting point for the modeling, but beidellite is not a phase typically observed in deep shale systems. And I have yet come across a shale (mudstone) with 42% microcline; this level of feldspar plus 50% clay would make this a very unusual rock.*

We thank the Reviewer and highlight that the methodology presented in not linked to the specific mineral assemblages or CCRs considered in this work. The choice of including a suite of three CCRs and different mineral assemblages (which maximize the generation of $CO_2$) is aimed at showing that our probabilistic workflow is flexible and can be readily adapted to include any mineral assemblage and CCR of interest. This is now clearly discussed in the revised Section 3.

*There are few things I am concerned about with respect to the reactions they picked. These represent just a very small number of possible reactions that*

*could take place during burial. So why not define the starting mineralogy, initi-
ate the burial process of increasing P and T, and let thermodynamics drive the
water-rock interactions to the most favored stable reactions. A priori selection
of the reactions seems rather narrow in thinking, although I do appreciate, they
wanted to target the most optimum reactions to produce CO2 but is this truly
"realistic".*

We apologize if the use of the expression *realistic* in the original manuscript
has generated some misunderstanding about the target of our work. Our intent
was to indicate that the system we consider is subject to conditions associ-
ated with space-time history of pressure and temperature which is consistent
with what one can observe in a sedimentary basin and characterize through a
burial model. We clarify this point in the revised manuscript (particularly, Ab-
stract, Introduction, and Section 2) by specifying that the expression realistic
sedimentary basin is employed to denote the three-dimensional evolution and
distribution of environmental pressure and temperature. We agree with the ob-
servation of the Reviewer about the variety of possible reactions taking place
in a complex geochemical setting of the kind associated with a sedimentary
basin. However, we note that considering an increased complexity system of
reactions (relying, e.g., on databases typically included in geochemical models)
would require including also a larger collection of uncertain parameters, thus
rendering the problem hardly tractable, at least at the current stage of develop-
ment. In the context of uncertainty quantification associated with geochemical
processes, a simplification of the underlying conceptual model is required, this
being an appropriate choice as long as the simplified model is still able to cap-
ture the key traits of the evolution of the main target output variables of interest.
We analyzed the consistency of the outputs of our simplified conceptual model
with the results proposed by a widely used geochemical software (i.e. Phreeqc;
Parkhurst and Appelo, 2013) under various conditions of pressure and temperature. For completeness, we illustrate in the following the results obtained for reaction CCR1 which can also be found in the Electronic Annexes of Ceriotti et al. (2017). We selected the temperature (T) and pressure (P) combinations observed in our sedimentary basin showcase listed in Table 1 together with depths at which these conditions are found. Note that this analysis is confined to temperature values below 300 °C, higher temperature values being outside the range of applicability of the Phreeqc software and Thermoddem database. The software Phreeqc is then used to simulate the geochemical system and evaluate aqueous and gaseous speciation when mineralogical phases associated with CCR1 are considered (see Figure 1 depicting the corresponding screenshot from the Phreeqc code).

| Depth [m ] | P [bar] | T [ ° C ] |
|---|---|---|
| 1421 | 197.00 | 99..00 |
| 1970 | 251.00 | 121.40 |
| 2504 | 303.00 | 141.00 |
| 3071 | 359.00 | 160.90 |
| 3661 | 416.80 | 180.30 |
| 4301 | 479.00 | 200.00 |
| 5004 | 548.50 | 222.00 |
| 5621 | 609.00 | 240.90 |
| 6310 | 679.40 | 261.00 |
| 6919 | 736.10 | 280.50 |

**Table 1.** Values of pressure (P) and temperature (T) resulting from the burial model at diverse depths at time t = 0 Ma.

Figure R.2 depicts (in logarithmic scale) the activities and molalities of these dissolved species as a function of depth. Results from Phreeqc are juxtaposed to the probability density function (denoted as $f_{C,Z}$) of CO2 fugacity resulting from our simplified model in the same conditions. We consider the results of

[Figure]

**Fig. 1.** Code implemented in Phreeqc to compute molalities and activities of the diverse aqueous species contributing to the molality/activity of dissolved $C(4)$ (i.e., $CO_{2(aq)}$; $HCO_{3(aq)}^-$; $Ca(HCO_3)_{(aq)}^+$; $CaCO_{3(aq)}$; $CO_3^{2-}$, and $MgCO_{3(aq)}$) for the values of temperature and pressure listed in Table 1.

our simplified geochemical system fully consistent with those obtained upon relying on the simulation of a complex network of reactions.

[Figure]

**Fig. 2.** Activities and molalities of the aqueous species contributing to the molality/activity of dissolved C(4) as computed through the Phreeqc software relying on the Thermoddem database compared to CO2 fugacity yielded by our probabilistic modeling.

*Second, they consider a system where the fluid is pure water which is very unrealistic when is it comes to sedimentary basin fluids; most are saline (typically 50-100 g/kg TDS). This would change the activity of water and in turn impact the solubility of the CO2 (the salting-out effect). High concentrations of CO2 would also affect the activity water. And to be sure a different activity of water*

*would impact the nature of the reactions they did identify.*

As pointed out by the Reviewer, the ionic strength of the brine may have a marked impact on the geochemical behavior of $CO_2$, as dissolved or gaseous phase.

As also acknowledged by the Reviewer, salinity of the brine may vary broadly in sedimentary basins. This issue should be approached in the context of a probabilistic assessment, in agreement with the main concept underlying our work.

Indeed, the selection of a certain level of salinity of the brine is an imposed initial condition, which can be as well subject to uncertainty due to our incomplete knowledge of the processes involved and other initial/boundary conditions in sedimentary environments. In this context, the choice of pure water is just one of the possible modeling scenarios. We agree that different (and perhaps more realistic) scenarios may be formulated, upon relying, e.g., on available information, which may be available from modeling or field observations on a specifically targeted sedimentary system. However, we remark that our study is focused on the quantification of parametric uncertainty and assumes a selected model structure which does not consider ionic strength. The motivation underlying our choice is now clarified in the revised Section 3, as reported in the following.

"Our study relies on a given model structure, thus neglecting uncertainty in the latter. We rest on the equilibrium-based approach used by Ceriotti et al. (2017). Thus, we consider pure mineral phases while neglecting other factors which would eventually influence the model structure (e.g., the occurrence of other mineral transformations, or effects associated with salinity of brine). Consistent with this model structure, we consider the equilibrium constant of speciation reactions as the key source of parametric uncertainty. We note that this choice is motivated by the observation that temperature and pressure values observed in sedimentary systems lie outside the range of conditions where thermodynamic equilibrium constants are usually characterized (Blanc, 2012). In addition to parametric uncertainty, we also consider input uncertainty, defined as the uncertainty related to the description of the system (Walker, 2003), i.e., we assume that diverse CCRs may take place depending on the mineralogical assemblage. These two sources of uncertainty are propagated throughout the final modeling goals of interest, i.e., the $CO_2$ source location, the $CO_2$ generation rate, and the temperature and pressure of CCR activation."]

*Third, they seem to pull thermodynamic data from what I consider outdated references. For example, Ian Hutcheon's work is certainly respected, but the authors should be very careful using thermodynamic data/insights that date back over 20 years. I recommend the authors take a journey through some the sources provided here (and associated references) just to be sure they are on the right path (this falls into the category of capturing "thermodynamicuncertainty"): Modeling Metamorphic Rocks Using Equilibrium Thermodynamics and Internally Consistent Databases: Past Achievements, Problems and Perspectives Pierre Lanari, Erik Duesterhoeft Journal of Petrology, Volume 60, Issue 1, January 2019, p. 19-56 https://doi.org/10.1093/petrology/egy105 CHNOSZ: Thermodynamic Calculations and Diagrams for Geochemistry Jeffrey M. Dick Front. Earth Sci., 16 July 2019 https://doi.org/10.3389/feart.2019.00180 Thermodynamic Data for Geochemical Modeling of Carbonate Reactions Associated with $CO_2$ Sequestration – Literature Review (only focuses on carbonates but still may be useful) KM Krupka KJ Cantrell BP McGrail: September 2010 Prepared for the U.S. Department of Energy under Contract DE-AC05-76RL01830 Qualification of Thermodynamic Data for Geochemical Modeling of Mineral-Water Interactions in Dilute Systems T. J.Wolery and C. Jove-Colon ANL-WIS-GS-000003 REV 00 November 2004 Zimmer, K., Zhang, Y.L., Lu, P., Chen, Y.Y., Zhang, G.R., Dalkilic, M. and Zhu, C. (2016) SUPCRTBL: A revised*

*and extended thermodynamic dataset and software package of SUPCRT92. Computer and Geosciences 90:97-111.doi.org/10.1016/j.cageo.2016.02.013. Rating: Good (leaning toward Fair)*

We thank the Reviewer for the constructive comment and for bringing these relevant references to our attention. While the works of Hutcheon are seminal in the context of CCR geochemistry, they do not constitute the source of the thermodynamic data used in our study. Thermodynamic data are taken form the Thermoddem database (Blanc et al., 2012) due to its completeness, traceability of data, and proven internal thermodynamic consistence, especially for the aluminum silicate phases (Blanc et al., 2015). We now detail all data sources in the manuscript by including the aforesaid literature references in Section 3.1 and in the Supplementary Material where thermodynamic data of all phases included in the work are listed in Table S1. Additional details on the procedure for estimating the uncertainties associated with thermodynamic constants starting from raw data are also included in the revised version of the manuscript (as Supplementary Material) to clarify the source of these information.

*Presentation quality: Are the scientific results and conclusions presented in a clear, concise, and well-structured way (number and quality of figures/tables, appropriate use of English language)?*
*Given the approach they adopted, the results and conclusions seem reasonable. But that said, this does not make them necessarily significant or innovative. They took an interesting approach with their probabilistic modeling then seemed to engineer the outcome they wanted by orchestrating a narrow type of reaction chemistry rather than truly allowing a more realistic mineralogic system to evolve during burial. I appreciate the probabilistic approach differs significantly from say a full developed 3-D reactive transport model that allows the system to track the thermodynamically most favorable reactions with accom-*

*panying fluid evolution. And this of course only considers the system from an equilibrium thermodynamic point of view.*

*Although more difficult, one could also address the evolution of the model system by quantitively assessing where and when the system deviates from equilibrium during burial. I guess I was hoping for better articulation of the connection between the evolution of the mineralogy and CO2 and the changes in porosity-permeability. For example, was CO2-rich fluid allowed to migrate from one unit to another or was every rock unit treated as a closed system.*

*The probabilistic approach is certainly interesting but seems to fail in capturing the dynamics of a complex heterogenous system undergoing change over non-trivial length and time scales.*

*So, what did we learn from this paper? We learned that if you take a dolomite-bearing or dolomite-rich rock containing other phases like some clay or feldspar and push the rock to higher P and T approaching low-grade metamorphism in a deep basin you can make lots of CO2. I think we kind of already knew this. What would have been really interesting to see is how this CO2 concentration changed as burial proceeded for each rock unit as a function of space and time, and how these changes affected the porosity and permeability of each unit. I am thinking they may have such information and if so, I encourage them to expand their outcomes to be more inclusive.*

*Rating: Good*

We agree with the Reviewer about the observation that our model does not include a high number of processes and is a streamlined representation of the geochemical burial complexity. Considering equilibrium rests on the hypothesis that the water-rock system located at a certain depth attains equilibrium before being buried to a deeper level. Thus, it is the burial velocity that limits the rate in the generation of CO2. This is indeed implicitly embedded in Equation (6) of the manuscript and fully acknowledge in Ceriotti et al. (2017). For completeness and clarity, we now explicitly state this element in the revised manuscript when presenting Equation (6) (lines 230 - 245).

We further emphasize that our work is not aimed at a comprehensive description of all bio-geochemical, geological, and fluid dynamics processes taking place in a sedimentary basin during diagenesis. The scope of the study is to present and apply a methodological framework and workflow for the probabilistic quantification of CO2 generation sources. As mentioned above and in the response to Reviewer 1, we revised parts of the Abstract, Introduction, and Conclusions to unambiguously clarify this element and avoid misleading terminology which might lead to expectations of the formulation of a full biogeochemical and geological model.

**1   References**

Battistelli A., Berry P., Bondua' S., Bortolotti V., Consonni A., Cormio C., Geloni C. and Vasini E. M. (2016) Thermodynamics-related processes during the migration of acid gases and methane in deep sedimentary formations. Greenhouse Gases, 7:295–312.

Blanc P., Lassin A., Piantone P., Azaroual M., Jacquemet N., Fabbri A. and Gaucher E. C. (2012) Thermoddem: a geochemical database focused on low temperature water/rock interactions and waste materials. Applied Geochemistry 27(10), 2107-2116.

Blanc P., Vieillard P., Gailhanou H., Gaboreau S., Gaucher E'., Fialips C. I., Made' B. and Giffaut E. (2015) A generalized model for predicting the

thermodynamic properties of clay minerals. Am. J. S. 315(8), 734-780.

Ceriotti G., Porta G., Geloni C., Dalla Rosa M., and Guadagnini A. (2017) Quantification of CO2 generation in sedimentary basins through carbonate/clays reactions with uncertain thermodynamic parameters, Geochimica et Cosmochimica Acta, 213, 198-215.

Menafoglio, A., Guadagnini, A., and Secchi, P.: Stochastic simulation of soil particle-size curves in heterogeneous aquifer systems through a Bayes space approach, Water Resources Research, 52, 5708-5726, 2016.

Neuman S. P. (2003) Maximum likelihood Bayesian averaging of uncertain model predictions. Stochastic Environmental Research and Risk Assessment. 17, 291-305.

Parkhurst D. L. and Appelo C. (2013) Description of input and examples for PHREEQC version 3 - a computer program for speciation, batch-reaction, one-dimensional transport, and inverse geochemical calculations. US Geological Survey Techniques and Methods, Book 6, 497.

Walker W.E , Harremoes P , Rotmans J. , Van Der Sluijs J.P. , Van Asselt M.B.A. , Janssen P. And Krayer Von Krauss M.P. (2003) Defining Uncertainty A Conceptual Basis for Uncertainty Management in Model-Based Decision Support, Integrated Assessment, 4(1), 5 - 17.

---

## Author Comment (AC3) · 14 Mar 2021

Dear Reviewer:

We appreciate the efforts you have invested in reviewing our manuscript. We are now providing our responses to the comments received for your

consideration. In the following reviewer comments are in italic, our responses
are in plain text, proposed changes in blue.

Sincerely,

Giulia Ceriotti (on behalf of all the authors)
*This manuscript provides a three-dimensional test case study to assess the
generation of CO2 by carbonate/clay reactions in a statistical framework. The
authors extend the work of Ceriotti et al (2017) to a synthetic field setting. The
article provides nice insights about the generation and source location of CO2.*

We thank the Reviewer for the constructive review and comments.
*I think that the article deserves publication after some minor corrections that
should address several points:*
*(1) the authors should take some time to explain the source of uncertainty in
equilibrium constants. Explain why the coefficients associated with equilibrium
constants are random and which is the underlying process. Also, it is important
to discuss the parameters describing this uncertainty from a realistic point of
view. Are these values realistic? Do they compare with a real field?*

We have revised the text to provide more context to our choices and with refer-
ence to the main elements associated with modeling and propagation of uncer-
tainty. As mentioned also in our response to Reviewers 1 and 2, we consider
uncertainty associated with equilibrium thermodynamic constants because our
application requires to extrapolate these beyond the temperature and pressure
conditions at which they are typically evaluated. The probability distribution
functions associated with these uncertain parameters are taken from our pre-
vious work Ceriotti et al. (2017). We have now amended the Supplementary
Material and provided more details about these aspects.

*It is not clear why the authors state that equilibrium constant Ks,ph depends on pressure and temperature but equation (1) only has temperature and is valid only for p=1 bar;*

The effect of pressure is taken into account by Equation (3), following classical approaches in equilibrium geochemistry.
*(2) some of the assumptions are not clear but from the introduction I was expecting some two-phase flow simulations besides generation;*

We have revised the Abstract and Introduction to sharpen the description of the objectives of our work and avoid misunderstandings. The objectives are now stated as follows in the Introduction:
"Modeling of $CO_2$ generation and accumulation in large-scale geological systems is typically prone to considerable uncertainties, chiefly due to paucity of information and to the large spatial and temporal scales involved. In this context, we provide a modeling framework that leads to a probabilistic quantification of the generation of $CO_2$ by a specific class of reactive processes (i.e., CCRs). As such, our study fills a knowledge gap by providing a methodology to support quantitative investigations of spontaneous $CO_2$ generation in large scale geological systems, these being otherwise typically based on mostly qualitative analyses. While we consider a simple geochemical model based on thermodynamic equilibrium, our probabilistic framework of analysis is flexible and can include treatment of model uncertainty (Walker, 2003; Neuman, 2003) as an additional element. Setting a given model structure is simply a convenient choice to minimize computational and conceptual complexity while at the same time considering a mathematical model that can be characterized with information that is typically available in field scale settings (in terms of, e.g., mineral composition, pressure, and temperature distributions). Values of equilibrium constants are here considered as uncertain because temperature and

pressure values observed in sedimentary systems lie outside the range of conditions where these parameters are usually characterized (Ceriotti et al., 2017; Blanc, 2012). In this work we investigate the propagation of this parametric uncertainty in the presence of various (alternative) CCR formulations by focusing on a three-dimensional scenario. When considering the framework proposed by Walker et al. (2003), our work allows combining uncertainty in model parameters (equilibrium thermodynamic constants) with input uncertainty, i.e., uncertainty in the description of the reference system. The latter type of uncertainty is reflected by our choice of considering diverse mineral assemblages leading to the occurrence of different CCRs. Note that our approach is geared towards quantification on the space-time location and intensity of the $CO_2$ source. This information can then be used as input to quantify scenario uncertainties, by delineating the spatial and temporal extent of $CO_2$ influx. Transport and accumulation of $CO_2$ across the subsurface can then be analyzed through approaches such as those described, e.g., in Battistelli et al. (2016). From an operational standpoint, our approach could be applied to enhance our knowledge on the degree of compatibility of $CO_2$ concentrations observed in field scale systems with the occurrence of CCR, as opposed to the action of other processes which might be considered in a large scale transport model of choice. The study is structured as follows (...)"

*(3) the authors state too many times that something can be found in Ceriotti et al (2017), this makes the description of methods not self-contained;*

Prompted by the Reviewer's concern, we have provided more context to the methodology in the Supplementary Material. Here, we include an outlined description of the procedure used to derive the statistics of the uncertain thermodynamic parameters starting from raw data included in Blanc et al. (2012).

*(4) not clear what do you mean by phases in equation (2) and reaction CCR, may be this is too abstract, and the authors should give an example to follow;*

We replace the wording "phase" with the more explicit "mineral phases" when needed in Sections 3 and 4.

*(5) sorry if I miss this one but I do not see the definition of $C_A$, not clear how is it calculated.*

$C_A$ is the frequency of activation of a CCR at a given spatial location (x, y, z). We have rephrased its definition at line 260.

*Other than this, I think that the manuscript is well written and organized and will be a nice contribution for HESS. I did not see typos except in line 162 where you are missing "it is".*

Thank you, we fixed the typo.

**1   References**

Battistelli A., Berry P., Bondua' S., Bortolotti V., Consonni A., Cormio C., Geloni C. and Vasini E. M. (2016) Thermodynamics-related processes during the migration of acid gases and methane in deep sedimentary formations. Greenhouse Gases, 7:295–312.

Blanc P., Lassin A., Piantone P., Azaroual M., Jacquemet N., Fabbri A. and Gaucher E. C. (2012) Thermoddem: a geochemical database focused on low temperature water/rock interactions and waste materials. Applied Geochemistry 27(10), 2107-2116.

Blanc P., Vieillard P., Gailhanou H., Gaboreau S., Gaucher E'., Fialips C. I., Made' B. and Giffaut E. (2015) A generalized model for predicting the

thermodynamic properties of clay minerals. Am. J. S. 315(8), 734-780.

Ceriotti G., Porta G., Geloni C., Dalla Rosa M., and Guadagnini A. (2017) Quantification of CO2 generation in sedimentary basins through carbonate/clays reactions with uncertain thermodynamic parameters, Geochimica et Cosmochimica Acta, 213, 198-215.

Menafoglio, A., Guadagnini, A., and Secchi, P.: Stochastic simulation of soil particle-size curves in heterogeneous aquifer systems through a Bayes space approach, Water Resources Research, 52, 5708-5726, 2016.

Neuman S. P. (2003) Maximum likelihood Bayesian averaging of uncertain model predictions. Stochastic Environmental Research and Risk Assessment. 17, 291-305.

Parkhurst D. L. and Appelo C. (2013) Description of input and examples for PHREEQC version 3 - a computer program for speciation, batch-reaction, one-dimensional transport, and inverse geochemical calculations. US Geological Survey Techniques and Methods, Book 6, 497.

Walker W.E , Harremoes P , Rotmans J. , Van Der Sluijs J.P. , Van Asselt M.B.A. , Janssen P. And Krayer Von Krauss M.P. (2003) Defining Uncertainty A Conceptual Basis for Uncertainty Management in Model-Based Decision Support, Integrated Assessment, 4(1), 5 - 17.